# Aerosol light absorption and the role of extremely low volatility organic compounds

Antonios Tasoglou[1,5], Evangelos Louvaris[2,3], Kalliopi Florou[2,3] Aikaterini Liangou[2,3], Eleni Karnezi[1] Christos Kaltsonoudis[4], Ningxin Wang[1], Spyros N. Pandis[1,2,3]

[1]Department of Chemical Engineering, Carnegie Mellon University, Pittsburgh
[2]Department of Chemical Engineering, University of Patras, Patras, Greece
[3] Institute of Chemical Engineering Sciences (ICE-HT), FORTH, Patras, Greece
[4]Department of Mechanical Engineering, Carnegie Mellon University, Pittsburgh, United States
[5]RJ Lee Group, Inc., Monroeville, United States

## Abstract

A month-long set of summertime measurements in a remote area in the Mediterranean is used to quantify aerosol absorption and the role of black and brown carbon. The suite of instruments included a high-resolution Aerosol Mass Spectrometer (HR-ToF-AMS), and a Scanning Mobility Particle Sizer (SMPS) both coupled to a thermodenuder and an aethalometer, a photoacoustic extinctiometer (PAX$_{405}$), a Multi-Angle Absorption Photometer (MAAP), and a Single Particle Soot Photometer (SP2).

The average refractory black carbon (rBC) concentration during the campaign was 0.14 μg m$^{-3}$, representing 3% of the fine aerosol mass. The measured light absorption was two or more times higher than that of fresh black carbon (BC). Mie theory indicated that the absorption enhancement due to the coating of BC cores by non-refractory material could explain only part of this absorption enhancement. The role of brown carbon (BrC) and other non-BC light-absorbing material was then investigated. A good correlation (R$^2$=0.76) between the unexplained absorption and the concentration of extremely low volatility organic compounds (ELVOCs) mass was found.

## 1. Introduction

Atmospheric aerosol may influence climate in two ways: directly through scattering and absorbing radiation, and indirectly through acting as cloud condensation nuclei (IPCC, 2011) Black carbon is the dominant light absorbing aerosol component. BC is a distinct type of carbonaceous material that is formed mainly during combustion processes. In addition, some

organic aerosol (OA) has the ability to absorb sunlight. The OA with strong light absorption is called brown carbon (BrC) (Andreae et al., 2006).

The absorption of BC depends on its mixing state (Liu et al., 2015, 2017). Usually, BC is coated with scattering material causing its light absorption to increase due to the lensing effect (Fuller et al., 1999; Jacobson, 2001; Bond et al., 2006; Lack and Cappa, 2010). The absorption enhancement ($E_{abs}$) is defined as the ratio of the aerosol absorption coefficient ($b_{abs}$) over the $b_{abs}$ of the pure BC particles. The $E_{abs}$ accounts for the combined effects of lensing and the presence of BrC. This enhancement can also be calculated by the ratio of the equivalent mass absorption cross-sections (MAC). The MAC is defined as the ratio of $b_{abs}$ over the BC mass. The MAC of pure, uncoated BC at a specific wavelength depends both on the size distribution of the particles, but also on their morphology (fresh aggregates versus collapsed more spherical structures). Forestieri et al. (2018) measured the size dependent MAC for particles with a diameter lower than 160 nm. Radney et al. (2014) and Dastanpour et al. (2017) showed that the MAC is proportional to particle mass. Based on Bond et al. (2006) pure BC particles in the 10-350 nm diameter range have an expected $MAC_{550}$ =3.9-6.8 m$^2$ g$^{-1}$ and $MAC_{405}$ =5.3-9.2 m$^2$ g$^{-1}$. The $E_{abs}$ can be measured by using a thermodenuder (TD) for the removal of the non-refractory coating material from the BC-containing particles or it can be estimated using theoretical models (for example Mie or Rayleigh-Debye-Gans theory) if the size of the primary spherules is known. Incomplete removal of the non-refractory material in the TD can lead to underestimation of the $E_{abs}$ (Healy et al., 2015; McMeeking et al., 2014).

Previous studies have demonstrated that lensing has a wide range of effects on the light absorption of BC. Liu et al. (2015) quantified the $E_{abs}$ in a rural area near London during the winter. Using the TD method, they found a campaign-average $E_{abs}$ of 1.3 at 405 nm and 1.4 at 781 nm. They also estimated the $E_{ab}$ combining their measurements with a reference MAC from the literature. At the large wavelength the two methods agreed, but this was not the case at the lower wavelength. In addition, they showed that there was continuous change in $E_{abs}$ with increasing $R_{BC}$. The $R_{BC}$ was defined as the ratio of the non-rBC mass to the rBC mass in BC-containing particles, measured by the SP-AMS. Finally, this study suggested that the lower volatility BrC had stronger absorption than the semi-volatile BrC. Knox et al. (2009) performed measurements in downtown Toronto, during the wintertime using a TD at 340 ºC. They reported an average $E_{abs}$ of 1.43 for fresh particles, based on thermal OC/EC and photoacoustic measurements. Liu et al. (2017)

combined laboratory experiments with diesel exhaust emissions and ambient measurements to show that particles with a ratio of $R_{BC}$ less than 1.5 (typical for traffic emissions) had a negligible lensing effect. The $R_{BC}$ was measured by combining an AMS and an SP2. When the $R_{BC}$ was above 3, lensing caused significant enhancement of the absorption of BC. They observed a continuous change in $E_{abs}$ with increasing $R_{BC}$. Zhang et al. (2018a) presented three years of measurements in a suburban site outside Paris, France, influenced by both fresh and aged air masses. On average they found an $E_{abs}$=2.07 at 370 nm, and an $E_{abs}$=1.53 at 880 nm. They calculated the $E_{abs}$ by measuring the absorption coefficient ($b_{abs}$) with an aethalometer, while the elemental carbon (EC) was measured with thermal methods using daily filters. Zhang et al. (2018b) presented measurements in Beijing, China during wintertime. Using aethalometer measurements, refractory and non-refractory particle size distributions and Mie theory, they calculated that the lensing effect led to an $E_{abs}$ ranging from 1.5 to 2 on average at 880 nm. Zanatta et al. (2018) found an average $E_{abs}$ equal to 1.54 at 550 nm for measurements at the Zeppelin Arctic Station. Lack et al. (2012) analyzed measurements of biomass burning plumes near Boulder, CO during summertime. Using a TD at 200 $^{o}$C, they found $E_{abs}$ values as large as 2.5 at 404 nm and 1.7 at 532 nm. Using the absorption Angström exponent (AAE) and Mie theory calculations they showed the presence of BrC.

Some studies have argued that Mie theory may overestimate the $E_{abs}$. Cappa et al. (2012) suggested that the absorption enhancement of BC in California in the summertime was low with average values equal to 1.13 at 405 nm and 1.06 at 532 nm for $R_{BC}$>10. For their measurements they used a TD operated at 225-250 $^{o}$C. Healy et al. (2015) also reported practically no enhancement in the BC absorption at 781 nm and an average $E_{abs}$=1.19 at 405 nm in Toronto, Canada during summertime. During a period associated with wildfires the same authors measured $R_{BC}$=6.9 and an average $E_{abs}$=1.39 at 405 nm. They argued that there was little evidence of the lensing effect, and that BrC was driving the $E_{abs}$. Cappa et al. (2019) performed measurements in Fresno, CA during wintertime and Fontana, CA during summertime. They found that in Fresno there was absorbing OA, BrC, which was related to biomass burning OA and nitrate-associated OA. In Fresno, they reported average $E_{abs}$ of 1.37, 1.22 and 1.1 at wavelengths equal to 405, 532, 781 nm, respectively, for $R_{BC}$ ranging from 1 to 4. In Fontana the $E_{abs}$ was lower with values of 1.1 at 405 nm and 1.07 at 532 nm. Laboratory measurements of McMeeking et al. (2014) for biomass burning aerosol indicated higher absorption in lower wavelengths compared to higher

ones and thus the presence of BrC. The enhancement of the absorption was negligible at $R_{BC}<10$.
On average they found an $E_{abs}$ equal to 1.25 at 781 nm with a maximum value of $E_{abs}=4$ for $R_{BC}>10$.
Recent studies have suggested that the absorption efficiency of OA could be related to its
volatility. Saleh et al. (2014) in their laboratory biomass burning experiments showed that almost
all absorbing OA was associated with extremely low volatility compounds (ELVOCs), with an
effective saturation concentration $C^*$ of $10^{-4}$ µg m$^{-3}$. In addition, Saleh et al. (2018) using controlled
combustion experiments showed that the absorption activity of BrC is proportional to its molecular
size.
Despite the significant progress in understanding the absorption of atmospheric fine aerosol
there are still remaining questions regarding both the absorption enhancement of black carbon and
the absorption of OA as the aerosol evolves in the atmosphere. In this study we try to address these
issues for aerosol that has been aged in the atmosphere for at least a few days before arriving at
the island of Crete in the Eastern Mediterranean.

**2. Experimental Methods**
A remote location in the Eastern Mediterranean was used for the study of the absorption
and volatility of aged carbonaceous aerosol. The area is characterized by intense photochemistry,
especially during the summer (Pikridas et al., 2010) and is affected by pollutants transferred from
continental Europe, Turkey, Greece and Africa (Mihalopoulos et al., 1997; Lelieveld et al., 2002;
Kalivitis et al., 2011; Bougiatioti et al., 2014). Previous measurements have shown that the OA
reaching the area is highly oxidized regardless of its origin (Hidlebrandt et al., 2010; 2011). Lee
et al. (2010) showed that these oxidized organic compounds have much lower volatility than fresh
SOA. Long-term measurements in the region have revealed relatively high light absorption and
scattering by aerosol during the summer (Kalivitis et al., 2011; Vrekoussis et al., 2005).
The FAME-16 field campaign took place from May 9 to June 2, 2016. Measurements were
conducted at the Finokalia Station (*35$^o$ 20' N, 25$^o$ 40' E*, 250 m asl), a remote site on the island of
Crete in Greece (Mihalopoulos et al., 1997). The nearest large city is Heraklion with 150,000
inhabitants located 50 km west of Finokalia (Kouvarakis et al., 2000). There are no local sources
near the station, allowing the investigation of aged OA from different source regions. During this
study, two Saharan dust events occurred from May 12 till May 15 and May 21 till May 22.
A Scanning Mobility Particle Sizer (SMPS, TSI classifier model 3080, CPC model 3775)
was used to measure the number and the size distribution of the particles. The aerosol flow was
set at 1 L min$^{-1}$ and the sheath flow at 5 L min$^{-1}$. The sampling time was 3 min.
The mass concentration and the chemical composition of the particles were monitored
using a High-Resolution Time-of-Flight Aerosol Mass Spectrometer (HR-ToF-AMS, Aerodyne
Research, Inc.). SQUIRREL 1.56D and PIKA v1.15D were used for the data analysis, while for
the elemental ratio calculations the improved ambient calculation approach of Canagaratna et al.
(2015) was used. The HR-ToF-AMS was operated in V-mode with a sample time of 3 min. The
collection efficiency of the HR-ToF-AMS was calculated using the algorithm of Kostenidou et al.
(2007). The average CE was 0.64±0.2. Positive matrix factorization (PMF) analysis (Lanz et al.,
2007; Paatero and Tapper, 1994; Ulbrich et al., 2009) was performed using as input the high
resolution OA mass spectra and the mass-to-charge ratios (*m/z*) from 12 to 200.
A Single Particle Soot Photometer (SP2, Droplet Measurement Technologies) was used to
measure the BC size distribution and concentration. The SP2 had 8-channels and included a 1064
nm Nd:YAG laser  operating at 4 V and 3400 A. The instrument was calibrated using fullerene
soot (Alfa Aesar, stock 40971, lot L20W054, SSA=0.4) (Gysel et al., 2011). The calibration of the
SP2 was verified in separate experiments using a centrifugal particle mass analyzer (CPMA,
Cambustion). The SP2 mode mass and the CPMA mode mass were in good agreement with an
$R^2$=0.99 (Saliba et al., 2016). The data were analyzed using the Probe Analysis Package for Igor.
In our measurements, the number concentration of BC was low (<10,000 particles cm$^{-3}$), and thus
it was assumed that there were no coincidence artifacts in our measurements. The scattering
measurement was calibrated using monodisperse polystyrene latex (PSL) spheres. The BC number
concentration distributions measured by the SP2 were fitted using a log-normal  distribution to
account for particles smaller than the SP2 detection limit of approximately 50 nm (Ditas et al.,
2018). This extrapolation resulted in an increase of the BC mass concentration by 3-8% (Figure
S9), therefore the uncertainty introduced by BC outside the measurement window was minor.
A photoacoustic extinctiometer (PAX, Droplet Measurement Techniques) with a blue (405
nm) laser was used to measure the absorption ($b_{abs}$) and the scattering ($b_{scat}$) coefficients. Fullerene
soot and PSL spheres were used to calibrate the absorption and scattering signals, respectively. An
activated carbon denuder was placed in front of the PAX to remove $NO_2$. The $b_{abs}$ measurement
by the PAX has an uncertainty of less than 10% (Nakayama et al., 2015). Furthermore, a seven
wavelength aethalometer (AE31, Magee Scientific) was used to measure the $b_{abs}$ at 370, 470, 520,
590, 660, 880 and 950 nm and to calculate the absorption Angstrom exponent (AAE), which
describes the wavelength dependence of the $b_{abs}$. The aethalometer measurements were corrected
for scattering and multiple scattering artifacts following Saleh et al. (2014) and Tasoglou et al.
(2017) using the corrections suggested by Weingartner et al. (2003) and Kirchstetter and Novakov
(2007). High relative humidity (>70%) can introduce measurement artifacts in the measurement
absorption coefficient by filter- based techniques or photoacoustic methods (Arnott et al. 2003). A
diffusion drier was used upstream of the optical measurements. The campaign average temperature
and relative humidity were 22±4 $^o$C and 53±19 %, respectively. The measurement station had a
temperature-control system maintaining the temperature at approximately 25 $^o$C.

The thermodenuder (TD) used in this study, was placed upstream of the HR-ToF-AMS and

the SMPS. The TD design was similar to that developed by An et al. (2007) and is described by
Louvaris et al. (2017). The TD was operated at temperatures ranging from 25 $^o$C to 400 $^o$C using
several temperature steps from 25 $^o$C to 200 $^o$C over several hours and then rapidly (in 20 min)
increasing its temperature to the 375-400 $^o$C to investigate the presence of ELVOCs. One complete
cycle from 25 to 400 $^o$C and back to 25 $^o$C lasted approximately 10 h. Sampling was alternated
between the ambient line and the TD line every 3 minutes using computer-controlled valves.
Changes in particle mass concentration, composition, and size due to evaporation in the TD were
measured by the HR-ToF-AMS and the SMPS resulting in thermograms of OA mass fraction
remaining (MFR) as a function of TD temperature. The OA MFR was calculated as the ratio of
organic mass concentration of a sample passing through the TD at time $t_i$ over the average mass
concentration of the ambient samples that passed through the bypass line at times $t_{i-1}$ and $t_{i+1}$. The
sample residence time in the centerline of the TD was 14 s at 25 $^o$C, corresponding to an average
residence time in the TD of 28 s. The MFR values were corrected for particle losses in the TD due
to diffusion and thermophoresis. To account for these losses, sample flow rate as well as size- and
temperature-dependent loss corrections were applied following Louvaris et al. (2017)
corresponding to the operating conditions during the campaign. Less than 20% of the particulate
matter was lost in the TD at temperatures up to 100$^o$C. The losses increased at higher temperatures.
and at 400 $^o$C approximately 50% of particles larger than 50 nm was lost. The uncertainty
introduced by the loss correction was approximately 20% (Gkatzelis et al., 2016; Louvaris et al.,
2017). The final step of the data analysis was to average the corrected for CE and TD losses MFR
data based on temperature bins of 10$^{o}$C. The MFR calculation assumes implicitly that the OA
concentration remains constant during the measurement period. To ensure that this condition is
satisfied, if two consecutive OA ambient mass concentrations differed by more than 25%, the
corresponding MFR was not included in the analysis. Also, in order to ensure that the temperature
was constant during the measurement, the absolute difference between the two samples had to be
less than 5$^{o}$C. If this difference for a TD sample was higher, then the sample was not included in
our analysis. The same approach was used also for the factors resulting from the PMF analysis of
the AMS spectra. However, in this case a minimum concentration threshold of 0.1 μg m$^{-3}$ was used
for the ambient concentrations together with the criterion of the stability of the ambient
concentrations during the sampling period. MFR values corresponding to concentrations of the
PMF factors below this threshold were not included in the dataset. Approximately 75% of the OA
samples satisfied all these constraints and were used in the analysis. The corresponding
percentages were 65% and 70% for the two identified PMF factors. In the present work the
complete datasets will be analyzed together, averaging the corresponding measurements. More
details regarding the data analysis and the sensitivity tests of the TD measurements are provided
in the supplementary information.

The concentrations of gas-phase pollutants were measured using a Proton-Transfer

Reaction Mass Spectrometer (PTR-QMS 500, Ionicon Analytik) and gas monitors. The PTR-MS
was calibrated with a standard gas mixture of VOCs. The concentration of O$_3$ was measured using
a continuous O$_3$ analyzer (Thermo Scientific, 49i) and the concentrations of nitrogen oxides were
measured using a NO/NO$_2$/NO$_x$ analyzer (Thermo Scientific, 42i-TL).

## 3. Theoretical Analysis Methods

The dynamic TD evaporation model of Riipinen et al. (2010) together with the uncertainty

estimation algorithm of Karnezi et al. (2014) were used for the determination of the OA volatility
distribution. Inputs for the model included the ambient OA concentration, the OA density
calculated by the algorithm proposed by Kostenidou et al. (2007), the initial average particle size,
TD temperature, the MFR values, and TD residence time. In the volatility basis set framework of
Donahue et al. (2006), the volatility distribution is represented with a range of logarithmically
spaced $C^*$ bins along a volatility axis. In this study 6 bins with variable mass fractions were chosen.
For this 6-bin solution, the best 2% of the mass fraction combinations with the lowest error were
used to estimate the average mass fraction along with their corresponding standard deviation for
the uncertainty of each bin. Additionally the parameters that affect indirectly the calculated
volatility such as the effective vaporization enthalpy ($\Delta H_{vap}$), and the effective accommodation
coefficient ($a_m$) were estimated following Karnezi et al. (2014). Additional information regarding
the data analysis of the TD measurements are in the supplemental information (sections S1 and
S2).

A Mie theory model based on the work of Bohren and Huffman (1983) was used to
calculate the theoretical MAC and $E_{abs}$ at 405 nm assuming a spherical core-shell morphology
(China et al., 2015). The assumption that all BC particles were coated and had obtained a relatively
spherical shape can be justified by the lack of any local sources and the fact that all particles
reaching the site had been heavily processed in the photochemically active summertime
atmosphere of the Eastern Mediterranean. PMF analysis of the OA AMS spectra did not show any
fresh emissions like HOA or BBOA and no acetonitrile was detected. FLEXPART analysis
confirmed that the air masses measured were transferred from long distance areas (30% continental
Greece and the Balkans, 13% Aegean, 24% Africa, 33% Italy/Sicily). The measurements
suggested a positive correlation (R=0.31) between the $MAC_{405}$ and the ratio of non-refractory $PM_1$
to rBC with an intercept of 12.1 $m^2$ $g^{-1}$ (Figure S10).

For the BC core we assumed a refractive index of the core $n_{rBC}$=1.85+0.71i (Bond et al.,
2006) and a density of 1.8 g $cm^{-3}$ (Mullins and Williams, 1987; Park et al., 2004; Wu et al., 1997).
A non-absorbing coating of the BC core was assumed, with a refractive index of $n_{OA}$ = 1.55 (Bond
and Bergstrom, 2006). The total aerosol effective density used was calculated based on the SMPS
and HR-ToF-AMS distributions. The average effective density for the campaign was 1.66±0.11 g
$cm^{-3}$. The size distribution of the BC cores was provided by the SP2 and the corresponding
measured rBC size distributions including the extrapolation to smaller sizes outside the SP2
measurement window.

The coating thickness in the base case calculation was estimated based on the assumption
that the BC material is internally mixed with the non-refractory aerosol species (Saliba et al.,
2016). The SMPS size distributions were used to estimate the ratio of the total aerosol mass over
the BC mass as measured. 1-hour averaged data were used as inputs in the model. In order to
provide a better constraint for the analysis, the average coating thickness for each period of the
corresponding mean BC cores (± 5 nm) was estimated using the leading-edge-only (LEO) fit
method (Gao et al. 2007) and the SP2 data. The thicknesses as expected were lower than those
resulting from the internal mixing assumption, but were still significant (Figure S11).  Mie theory
calculations using the coating thickness based on the LEO fit method were also performed.

**4.   Results and discussion**

The average rBC concentration of the campaign was 0.14 µg m$^{-3}$ and the average OA

concentration was 1.5 µg m$^{-3}$ (Figure 1). The two major Saharan dust events affected, as expected,
the aerosol optical properties. In the present study we focus only on the non-dust periods. The
dominant PM$_1$ components were sulfate and OA, accounting for 46% and 34% of the PM$_1$,
respectively. The O:C ranged from 0.65 to 1, with an average value of 0.83, revealing the absence
of fresh OA. These values are typical in Finokalia during the spring and summer periods
(Hidlebrandt et al., 2010).

**4.1 OA Volatility**

The estimated volatility distribution for the total OA in Finokalia during FAME-16 is

depicted in Figure 2. Use of OA with $C^* = 10^{-8}$ µg m$^{-3}$ was needed to capture the behavior of the
OA at 400 $^{\circ}$C. Almost 40% of the OA consisted of semi-volatile organic compounds (SVOCs),
35% of low volatility organic compounds (LVOCs), and the rest was extremely low volatility
organic compounds (ELVOCs).

The estimated value of the effective vaporization enthalpy was $80 \pm 20$ kJ mol$^{-1}$. This value

was in agreement with the reported value by Lee et al. (2010) of 80 kJ mol$^{-1}$ for the FAME-08
campaign. The estimated accommodation coefficient was 0.27, ranging from 0.1 to 0.8. This value
was a little higher than the 0.05 value reported in the earlier study. However, both suggest only
moderate resistances to mass transfer during the evaporation in the TD.

The corresponding measured and predicted thermograms are depicted in Figure 3. Almost

30% of the OA had not evaporated even after heating at 400 $^{\circ}$C. The temperature at which half of
the OA evaporated was T$_{50}$=120 $^{\circ}$C, a value similar to that observed by Lee et al. (2010). The
composition of the OA leaving the TD changed significantly as temperature increased according
to the model. At 125 $^{\circ}$C the LVOCs and ELVOCs contributed equally to the remaining OA mass.
For further temperature increases the LVOC fraction was reduced until 375 $^{\circ}$C, at which point only
the ELVOCs remained.
PMF analysis resulted in a two-factor solution (Florou et al., in prep.). Factor 1
corresponded to more oxidized oxygenated OA (MO-OOA) and Factor 2 to a less oxidized
component (LO-OOA). The average contribution of the two factors was 47% for the MO-OOA
and 53% for the LO-OOA.  The O:C for the MO-OOA was 0.95 ($OS_C$ = 0.59) and for the LO-
OOA it was 0.56 ($OS_C$ = -0.27). There was a weak positive correlation between LO-OOA, MO-
OOA and rBC. The $R^2$ between the hourly concentrations of MO-OOA and rBC was 0.12 (Figure
S12), and for LO-OOA and rBC 0.17 (Figure S13). This is not unexpected given that Finokalia is
far away from the corresponding sources of both BC and organic compounds and significant
physical and chemical processing has taken place during the transport of the aerosol from the
sources to the receptor.
The OA in the Eastern Mediterranean during the summer has significant contributions from
both anthropogenic and biogenic sources but their contributions remain uncertain. Based on our
recent work (Drosatou et al., 2019) the LO-OOA and MO-OOA do not reflect different sources,
but rather different degrees of chemical aging. The Finokalia area is characterized by the absence
of local sources and as result the absence of fresh OA. The area is characterized by dry land and
dry vegetation (dry bushes), thus no high biogenic emissions are expected. The OA during FAME-
08, was also found to consist entirely of OOA with no primary OA (POA) present (Hildebrandt et
al., 2010). POA evaporates and gets oxidized rapidly in the photochemically active environment
of the Eastern Mediterranean during its transport from its sources to this remote site. FLEXPART
simulations during the measurement campaign revealed that the air masses arriving at the site
during the study were originating from continental Greece and the Balkans (30% of the time),
Aegean (13% of the time), Africa (24%), and Italy/Sicily (33%).
The volatility distributions of the two PMF factors were estimated following the same
approach as that for the total OA. The measured thermograms for the two PMF factors are shown
in Figure 4. Almost 50% of both the MO-OOA and the LO-OOA evaporated at 150 °C. Almost
30% of the MO-OOA mass, and about 20% of the LO-OOA did not evaporate even at temperatures
as high as 400°C. The model reproduced the observed MFR values for both factors. Both factors
contained components with a wide volatility range. The MO-OOA exhibited a bimodal volatility
distribution with peaks at effective saturation concentrations of $10^{-8}$ and 10 µg m$^{-3}$. Its effective
enthalpy of vaporization was approximately $90 \pm 35$ kJ mol$^{-1}$ and its accommodation coefficient
was 0.27. The estimated volatility distribution of the LO-OOA was a little more uniform peaking

at an effective saturation concentration of 1 µg m$^{-3}$. The average calculated saturation concentration of LO-OOA at 298 K was 0.016 µg m$^{-3}$, an order of magnitude higher than that of the MO-OOA. The LO-OOA enthalpy of vaporization was 70 ± 20 kJ mol$^{-1}$, 20 kJ mol$^{-1}$ lower than that of the MO-OOA. Its accommodation coefficient was approximately 0.1 indicating small mass transfer resistances. MO-OOA consisted of approximately 40% SVOCs, 30% LVOCs, and 30% ELVOCs. On the contrary, LO-OOA consisted of almost 45% SVOCs, 40% LVOCs and only 15% ELVOCs. ELVOCs can be produced both by primary sources (combustion of fossil fuels but also biomass burning) and secondary processes. Given the intense chemical processing of the organic compounds from all sources in their way to Finokalia, we cannot assign based on our measurements to a specific source or process.

The fitting of the individual factor thermograms implicitly assumes that each factor had the same size distribution as the total OA and also that the two factors were externally mixed. The uncertainty introduced by these two assumptions was implicitly evaluated comparing the estimated total OA of volatility distribution with the composition-weighted average of the volatility distributions of the two OA factors. The two distributions agreed within a few percent for $10^{-3}<C^*<10^0$ and within 10% for the lowest and highest volatility bins.

**4.2 Aerosol optical properties**

The PAX was used to measure the $b_{abs}$ and $b_{scat}$ at 405 nm. The measured $b_{scat,405}$ ranged from 11.5 to 47 Mm$^{-1}$ with an average campaign value of 26.5 Mm$^{-1}$ (Figure 5). The corresponding $b_{abs,405}$ ranged from 0.64 to 5.6 Mm$^{-1}$ with an average value of 2.1 Mm$^{-1}$.

The absorption coefficients measured by the aethalometer at λ=370 nm and at λ=470 nm, were compared with the absorption coefficient measured by the PAX at λ=405 nm. The measurements of the two instruments were highly correlated with $R^2$=0.9-0.91. The $b_{abs,370}$ measured by the aethalometer was higher than the $b_{abs,405}$ of the PAX. Their relationship is described by the equation $y=3.32x$ - 3.24. Similarly, the $b_{abs,450}$ measured by the aethalometer was higher than that of the PAX at 405 nm and their relationship was described by $y=1.85x – 0.93$ (Figure S14). Part of these differences are due to the artifacts associated with filter-based absorption measurements (Cappa et al., 2008; Lack et al., 2008).

Acetonitrile is a known biomass burning marker and can help identify the potential influence of the site by biomass burning events or wildfires during the campaign. The acetonitrile

concentration measured by the PTR-MS remained close to 0.4 ppb during the campaign, which is the local background level. This together with the low BC levels indicate that the site was not impacted by nearby biomass burning during the study.

The $b_{abs,405}$ variation followed that of the rBC ($R^2$=0.74 for the hourly averages). The $R^2$ at λ=370 nm was lower at 0.67 than that at 405 nm. This is consistent with the presence of some BrC. However, the $b_{abs,370}$ is measured by the aethalometer and the $b_{abs,405}$ by the PAX. For the $b_{abs}$ at higher wavelengths measured by the aethalometer the $R^2$ at λ=450 nm was 0.68, at λ=520 nm 0.67, at λ=590 nm 0.65, at λ=660 nm 0.65, and at λ=950 nm 0.51 (Figure S15). These values are inconclusive regarding the existence or absence of BrC. Part of the explanation for this behavior could be that a fraction of the BrC is associated with rBC either from the emissions or from the associated gas-phase pollutants that react in their way to the site.

The ratio of the $b_{abs,405}$, measured by the PAX, over the rBC mass, was equal to 16±2.2 $m^2$ $g^{-1}$. In this study the average rBC size distribution had a number mode diameter of 67 nm and a mass mode diameter equal to 185 nm. The measured $MAC_{405}$ is clearly higher than the expected $MAC_{405}$ of uncoated BC particles (Bond et al., 2006). The difference between the measured and the reference value of $MAC_{405}$ can be due to the coating of BC by other PM components (lensing effect) and/or the existence of other absorbing material. These two potential explanations will be explored in the following paragraphs.

The AAE of the aerosol was calculated using a power-law fitting of the $b_{abs}$ measured by the aethalometer in all seven wavelengths (370, 470, 520, 590, 660, 880, 950 nm). The campaign average AAE was 0.97±0.22. Lack and Cappa (2010) suggested that an AAE>1.6 should confirm the presence of non-BC absorbing material, however an AAE<1.6 does not exclude its presence. The AAE of coated BC cores can deviate from the typical AAE=1 with values greater or lower than unity. Gyawali et al. (2009) have shown using a core-shell model and assuming spherical particles that for relatively small cores, the coating thickness increase leads to the increase of the AAE while for large cores, AAE is not affected by the changes of the coating thickness. Liu et al. (2018) showed that the AAE of coated BC is highly sensitive to the particle size distribution and demonstrated that AAE decreases as particle size increases. They showed that an AAE as low 0.8 is possible and demonstrated the importance of various parameters on the BC AAE and the potential problems introduced by assuming BC AAE as being equal to 1.0. Based on the relatively large aged particles present in Finokalia relatively low values of AAE should be expected.

According to simulation presented by Gyawali et al. (2009), the average AAE for this study can
range from 1 to 1.25, based on the core diameter measured and the estimated coating thickness.
Therefore, the relatively low average AAE does not preclude the presence of some absorbing
organic aerosol.
Mie theory calculations were performed in order to estimate the $MAC_{405}$ and the $E_{abs}$ due
to the lensing effect of the shell covering the BC core. In our base case the coating thickness was
estimated based on the ratio of the total aerosol mass over the BC mass (Saliba et al., 2016).
Initially, a non-absorbing shell was assumed. The predicted $MAC_{405}$ had an average value of 15.8
$m^2 g^{-1}$. The average $E_{abs}$ due to the lensing effect was 2.13. The predicted average $MAC_{405}$ using
the measured BC size distribution and assuming pure uncoated spherical particles was 7.4 $m^2 g^{-1}$
varying from 6.1 to 7.8 $m^2 g^{-1}$ during the study (hourly averages). This is consistent with the
expected 5.3-9.2 $m^2 g^{-1}$ for BC core diameters in the 10-350 nm size range (Bond et al. 2006). The
average predicted $MAC_{405}$ was lower than the average measured $MAC_{405}$ =16 $m^2 g^{-1}$ but the p-
value was equal to 0.079. Taking into account the measurement uncertainty, the measured average
$MAC_{405}$ could range from 14.4 to 17.6 $m^2 g^{-1}$. However, here were several periods during which
the measured MAC was much higher than the predicted values with the differences as high as 8.5
$m^2 g^{-1}$ (Figure 6). These differences suggest the average contribution of BrC to absorption during
the measurement period was relatively small, but there were periods with high levels of absorbing
OA.
In the next step, the Mie theory calculations were repeated assuming an absorbing shell
with a refractive index of $n_{OA} = 1.55 + ki$, where the imaginary part, $k$, was allowed to vary from
0 to 0.4. This range of $k$ values was selected based on previous literature (Kirchstetter et al., 2004;
Alexander et al., 2008; Chakrabarty et al., 2010; Chen and Bond, 2010; Saleh et al., 2014;
Chakrabarty et al., 2016, Li et al., 2016; Saleh et al., 2018). Approximately half of the resulting $k$
values during the campaign were zero, suggesting a non-absorbing shell, while the other half were
were positive. More specifically, 24% of the estimated $k$ values ranged from 0.01 to 0.1, 8% from
0.11 to 0.2, 3% from 0.21 to 0.3, and 10% of the $k$ values ranged from 0.31 to 0.4. During these
periods, the campaign average $b_{abs,405}$ was equal to 2.4 $Mm^{-1}$. BrC was estimated to lead to a 15%
increase of the campaign average $b_{abs,405}$.
The Mie calculations were repeated using a coating thickness estimated by the LEO fit
method and the SP2 data. The average coating thickness calculated by this approach was

approximately half of that calculated using the internal mixture assumption (Figure S11). This resulted in lower predicted absorption and therefore a larger gap between measurements and predictions assuming that the organic aerosol was not absorbing. The predicted $MAC_{405}$ had an average value of 14.4 $m^2\,g^{-1}$. The predicted average $MAC_{405}$ assuming pure uncoated spherical particles was 7.4 $m^2\,g^{-1}$. The $E_{abs}$ was equal to 1.94. Assuming an absorbing shell, we found that the imaginary part of the refractive index of the shell was 20% of the times in the range of 0.01 to 0.1, 16% between 0.11 to 0.2, 16% between 0.21 to 0.3, and 20% between 0.31 to 0.4. Only 28% of the times the $k$ was equal to 0 indicating no absorbing shell.

For both methods used to estimate the $MAC_{405}$ it is noted that there was presence of absorbing material in 45% up to 72% of the measurements.

**4.3 The role of ELVOCs**

ELVOCs can exhibit substantial larger light absorption than LVOCS or SVOCs. The association of photochemically aged BrC with material of lower volatility has been reported in a number of studies focusing on biomass burning (Saleh et al., 2014; Wong et al., 2019). The hypothesis that the presence of ELVOCs could explain the higher aerosol light absorption was tested. The unexplained MAC (ΔMAC) difference of measured and predicted values was compared with the total ELVOC mass concentration. The ELVOC concentration was estimated based on the results on the volatility analysis of the two PMF factors:

$$[ELVOC]= 0.15 [LO\text{-}OOA] + 0.3 [MO\text{-}OOA]$$

The predicted $MAC_{405}$ used was from the base case Mie theory calculations in which the coating thickness was estimated from BC mass fraction The unexplained $b_{abs}$ at 405 nm (3 h average values) was well correlated with the estimated ELVOC concentrations with an $R^2=0.76$ (Figure 7). Even after excluding the period with the highest unexplained absorption, the relationship is still relatively strong with $R^2=0.41$. The corresponding $R^2$ between the 3-h average ΔMAC and the ELVOCs was 0.66 (Figure S16). These results suggest that the ELVOCs were probably contributing to the total absorption and could explain the difference in the MAC.

Correlation of the ΔMAC with the rBC, and with other parameters were lower than those for the ELVOCs. For example, the $R^2$ with the rBC was 0.38, with the LO-OOA 0.44 and with the MO-OOA 0.29. There was a relatively high correlation with the sulfate levels ($R^2=0.59$) that could be interesting as high sulfate levels in this area correspond to high aerosol acidity which has been

shown to promote formation of oligomers in secondary organic aerosol. A comparison of the
difference of measured and the predicted $b_{abs,405}$ and the ELVOC was also conducted for all the
periods including those in which no BrC was present ($k=0$). The correlation was $R^2 = 0.52$. During
the $k=0$ periods the predicted $b_{abs,405}$ was equal or higher than the measured $b_{abs,405}$. This difference
was due to overestimation of the absorption values by the Mie theory or due to the uncertainties in
the measurements of the $b_{abs,405}$.

The calculations were repeated using the predicted MAC based on the Mie theory assuming

a coating thickness according to the LEO fit. The correlation between the unexplained absorption
and the ELVOC concentration remained high with $R^2=0.69$ (Figure S17), suggesting that despite
the uncertainty in the coating thickness our results are quite robust.

**5. Sensitivity analysis**

The analysis presented in the previous sections has been based on a series of assumptions

and, as expected is affected, by measurement uncertainties. We have tested the robustness of our
conclusion about the link between the unexplained absorption and ELVOC levels by repeating the
analysis for several cases. We focused on the estimation of the coating thickness of the BC
particles, the refractive index of black carbon, the uncertainty of the measurements by the SP2 and
$PAX_{405}$, the uncertainty of the results from the PMF analysis and finally the uncertainty of the
thermodenuder model. The results of the corresponding tests are summarized in the following
paragraphs.

The effect of the assumed BC refractive index on our results was tested by repeating the

Mie theory calculations for two additional values: a relatively high value of 1.95+0.79i and a
relatively low value 1.5 + 0.5i (Bond and Bergstrom, 2006). For the high refractive index, the
predicted average $MAC_{405}$ using the measured BC size distribution and assuming pure uncoated
spherical particles was 7.7 $m^2\,g^{-1}$ varying from 6.3 to 8.2 $m^2\,g^{-1}$ during the study (hourly averages).
Even if the unexplained absorption was reduced its $R^2$ with the ELVOCs remained high and equal
to 0.72 (Fig. S18). For the low refractive index, the predicted average $MAC_{405}$ of the uncoated BC
particles was 6.2 $m^2\,g^{-1}$ (range of 5.2-6.5 $m^2\,g^{-1}$ during the study).  The $R^2$ between the unexplained
absorption, $\Delta b_{abs,405}$ and the ELVOC concentration was $R^2=0.55$ (Fig. S19). The link between the
unexplained absorption and the abundance of ELVOCs is quite robust with respect to the assumed
value of the BC refractive index.
We estimated that 92-97% of the BC mass concentration was inside the SP2 measurement
window therefore given that we also corrected for it fitting the measured size distribution, the
uncertainty introduced by this limitation of the SP2 was minor. Nakayama et al. (2015) reported
an uncertainty of the $b_{abs}$ measurements by the PAX of less than 10%. Both of these uncertainties
did not have an important effect on the link between the unexplained absorption and the ELVOC
concentrations.
An assessment of the uncertainty of the concentrations of the two OA factors determined
by the PMF analysis was performed by bootstrapping 10 simulations (Ulbrich et al., 2009). The
estimated uncertainty for the LO-OOA concentrations was 2% and for the MO-OOA
concentrations was 3% (Fig. S20). These relatively small uncertainties suggest that the PMF
uncertainty regarding the determination of these factors does not affect significantly the
conclusions of this study.
The uncertainty related to the volatility distributions determined by the thermodenuder
results was assessed by estimating low and high limits of the ELVOC concentrations: $ELVOC_{low}$
and $ELVOC_{high}$. These were estimated based on the extreme mass fractions that were calculated
during the sensitivity analysis of the TD model (Supplemental information Section S2). The LO-
OOA ELVOC mass fraction ranged from 0.09 to 0.25 while that of the MO-OOA from 0.13 to
0.47. For the low ELVOC case we thus assumed that:

$[ELVOC_{low}]= 0.09 [LO\text{-}OOA] + 0.13 [MO\text{-}OOA]$

The $R^2$ between the unexplained $b_{abs,405}$ and $ELVOC_{low}$ was $R^2=0.79$ (Fig. S21). For the high
ELVOC case:

$[ELVOC_{high}]= 0.25 [LO\text{-}OOA] + 0.47 [MO\text{-}OOA]$

Once more the correlation between the unexplained absorption at 405 nm and the ELVOCs was
quite high with $R^2=0.78$ (Fig. S22).

## 6.  Conclusions

A month-long campaign was conducted at a remote site, in Finokalia, Crete during May of
2016. The dominant $PM_1$ components were sulfate and aged organics with O/C=0.81. The average
ambient OA concentration was 1.5 µg m$^{-3}$ and the rBC was 0.14 µg m$^{-3}$. Continuous monitoring of
biomass burning markers revealed that there were no periods of enhanced biomass burning
influence on the site during the campaign.  PMF analysis resulted in two secondary OA factors:

one more oxidized (MO-OOA) and one less oxidized (LO-OOA). Total OA consisted on average of 40% SVOCs, 35% LVOCs and 25% ELVOCs. Both OA components with a wide range of volatilities. Approximately 30% of the MO-OOA was ELVOCs, 30% LVOCs and 40% semi-volatile material. The LO-OOA was more volatile on average with 40% consisting of LVOCs and 45% of SVOCs and 15% of ELVOCs.

Aerosol optical properties were measured. The average $b_{scat}$ at 405 nm was 26.5 Mm$^{-1}$ and the average $b_{abs}$ at 405 nm was 2.1 Mm$^{-1}$. Furthermore, the average AAE was 0.97 and the MAC$_{405}$ was 15.7. Mie theory calculations were able to reproduce less than half of the measured MAC$_{405}$ values assuming core-shell morphology and a non-absorbing shell ($k$=0). We estimated that the non-absorbing shell was causing an enhancement of the absorption by a factor of 2.1. For the other half of the measurements the presence of an absorbing shell with an average $k$ of 0.18 was needed to explain the measurements.

The ELVOCs mass concentration was estimated using the volatility distributions of the two factors. The ELVOC concentration was highly correlated with the unexplained MAC$_{405}$ ($R^2$=0.66) and the unexplained $b_{abs,405}$ ($R^2$=0.76), defined as the differences of the parameters measured and the ones predicted by Mie theory for $k$=0. These results suggest that the unexplained absorption in this remote site could be due to a large extent to the extremely low volatility components of the organic aerosol.

*Data availability.* The data in the study are available from the authors upon request (spyros@chemeng.upatras.gr).

*Author contributions.* AT conducted the absorption measurements, analysed the results and wrote the paper. EL performed the thermodenuder measurements and analysed the results. KF performed the AMS measurements and analysed the results. AL performed the PTR-MS measurements and analysis. EK was responsible for the OA volatility analysis. CK coordinated the field campaign and assisted with all measurements. NW assisted with all measurements SNP was responsible for the design and coordination of the study and the synthesis of the results. All co-authors contributed to the writing of the manuscript.

*Competing interests.* The authors declare that they have no conflict of interest.

**Acknowledgements**
This work was supported by U.S. Environmental Protection Agency STAR program [grant
number R835035] and the project "Panhellenic infrastructure for atmospheric composition and
climate change, PANACEA" (MIS 5021516) which is implemented under the Action
"Reinforcement of the Research and Innovation Infrastructure", funded by the Operational
Programme" Competitiveness, Entrepreneurship and Innovation" (NSRF 2014–2020) and co-
financed by Greece and the European Union (European Regional Development Fund). Travel
support was provided by the European Research Infrastructure ACTRIS. The authors would like
to thank the Finokalia station personnel for the accommodation and for providing the aethalometer
measurements.

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

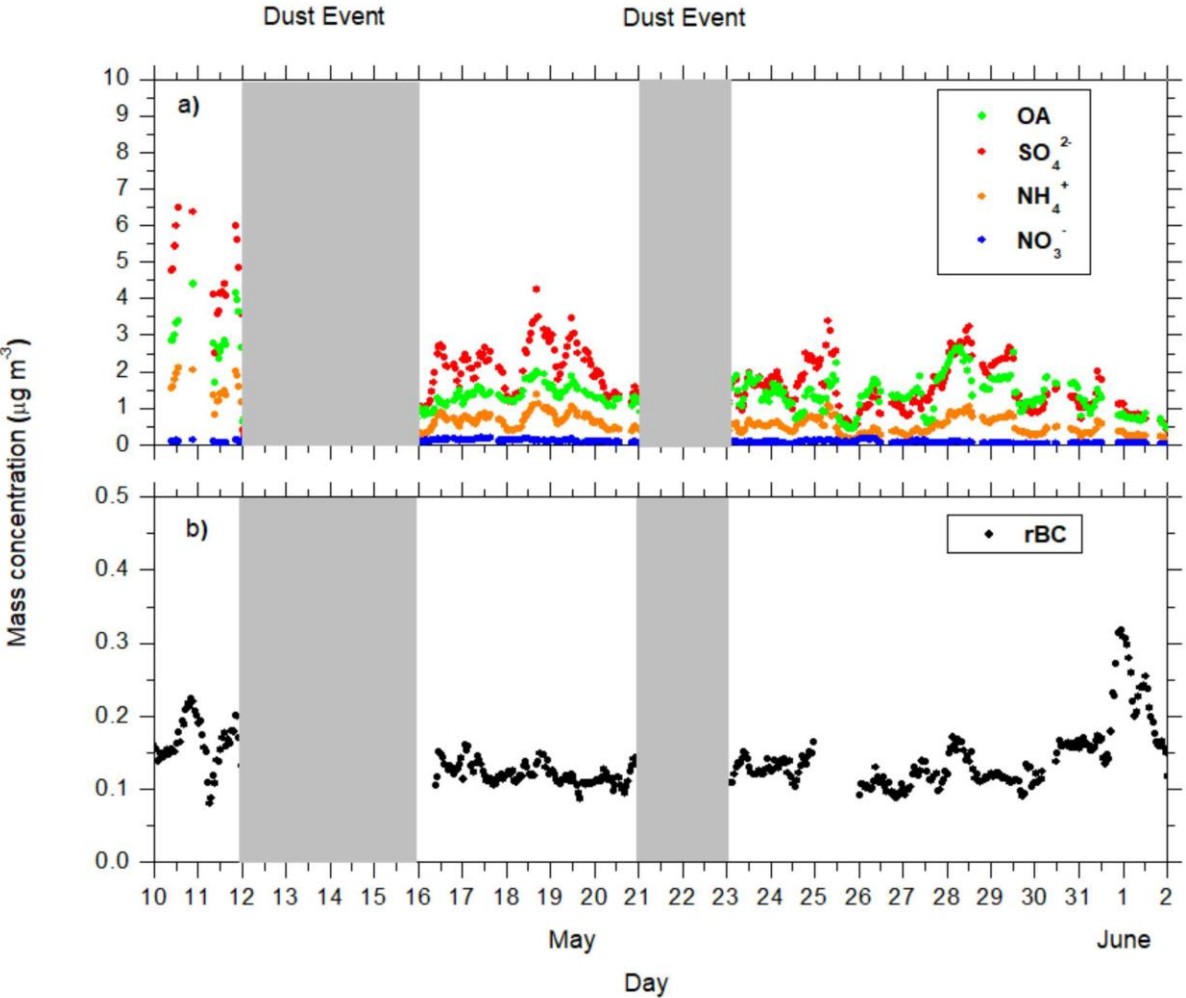


**Figure 1**: Evolution of the aerosol chemical composition based on the HR-TOF-AMS and SP2

measurements during FAME-16: a) the non-refractory aerosol components; b) rBC concentration.

The shaded areas represent the dust events periods.


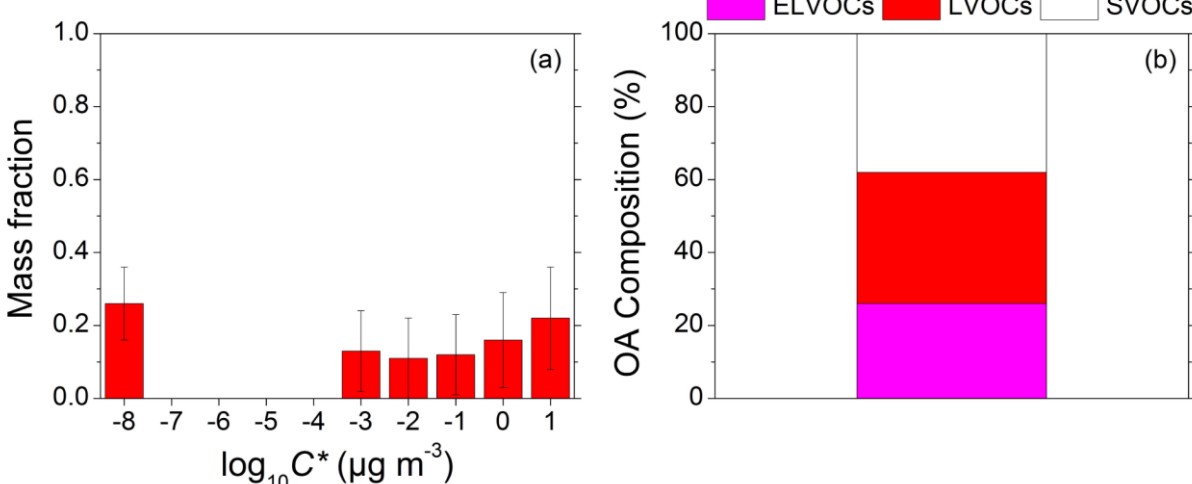


**Figure 2:** (a) Total OA volatility distribution along with its uncertainty estimated by the Karnezi et al. (2014) approach. The error bars represent the corresponding variability (± 1 standard deviation). (b) OA composition. Magenta color represents the ELVOCs, red the LVOCs, and white the SVOCs.

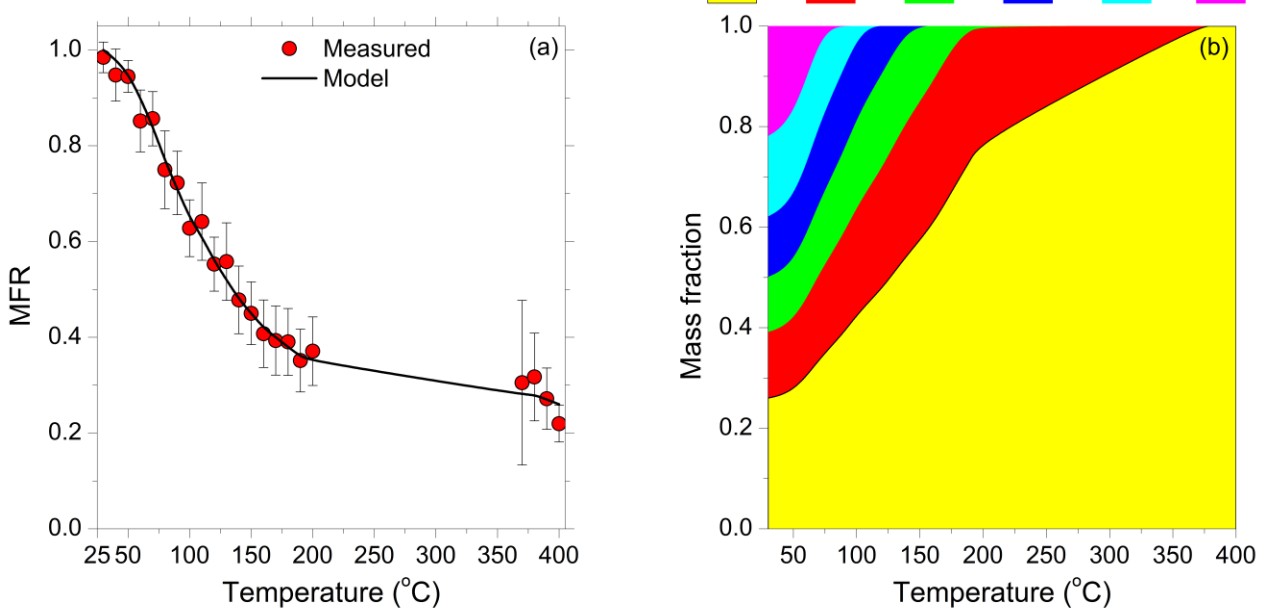

783

**Figure 3:** (a) Average loss-corrected total OA thermograms. Red circles represent the measured total OA MFR and the error bars the corresponding variability (±2 standard deviations of the mean). The solid lines are the model predictions (b) Mass fraction of the total OA for different effective saturation surrogate species with concentrations as a function of TD temperature. Yellow color represents the contribution of the effective saturation concentration $C^* = 10^{-8}$ μg m$^{-3}$, red the contribution of the $C^* = 10^{-3}$ μg m$^{-3}$, green the $C^* = 10^{-2}$ μg m$^{-3}$, blue the $C^* = 10^{-1}$ μg m$^{-3}$, cyan the $C^* = 10^0$ μg m$^{-3}$, and magenta the $C^* = 10$ μg m$^{-3}$.

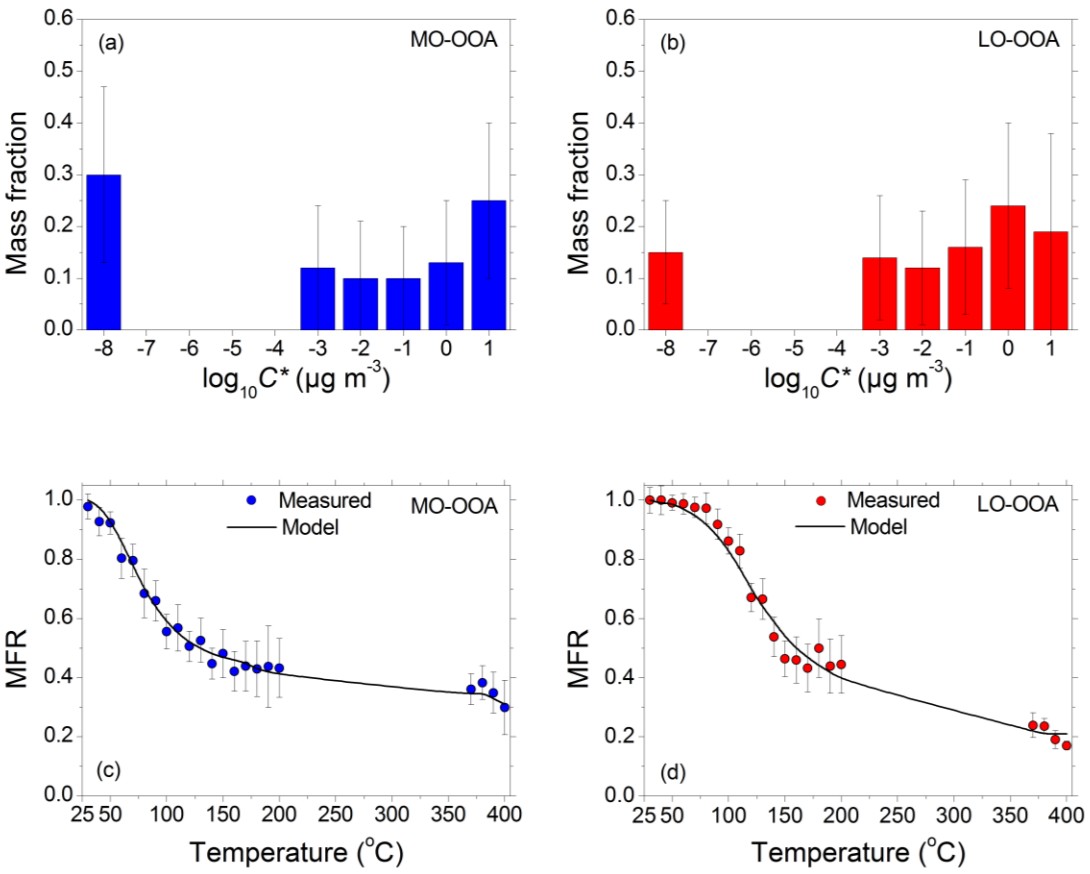

791

**Figure 4:** (a) Estimated volatility distribution of the MO-OOA factor along with its corresponding uncertainties by using the approach of Karnezi et al. (2014). (b) Estimated volatility distribution of the LO-OOA factor along with its corresponding uncertainties. (c) Measured (in circles) and predicted thermograms for the LO-OOA factor. The error bars represent ± 2 standard deviations of the mean. (d) Measured (in circles) and predicted thermograms for the LO-OOA factor. The error bars represent ± 2 standard deviations of the mean.

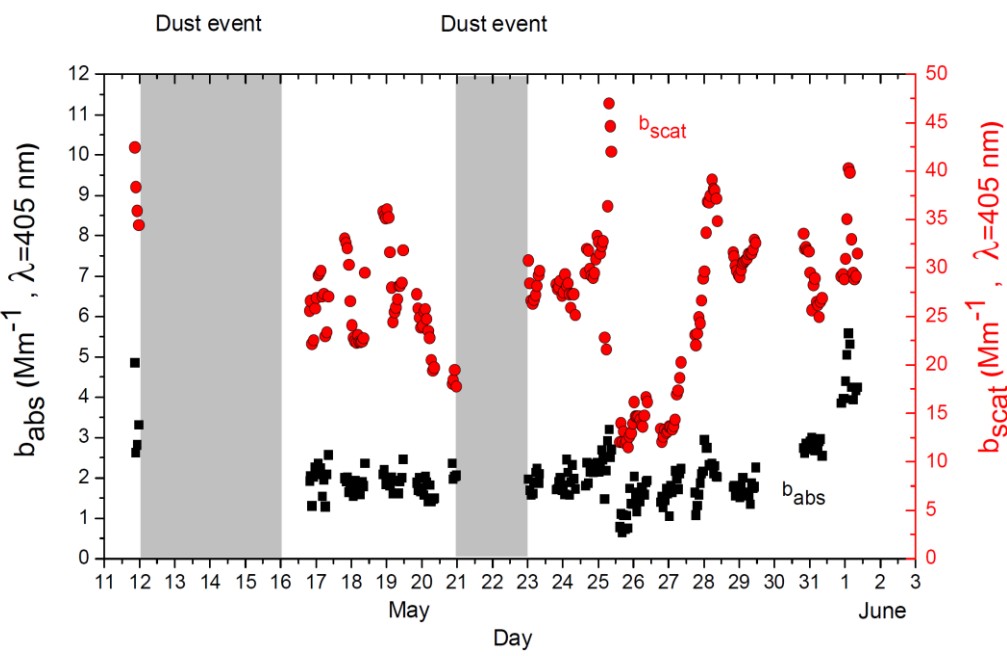


**Figure 5:** The timeseries of the aerosol optical properties at $\lambda$=405 nm. The black squares represent

the absorption coefficient, $b_{abs}$, while the red circles represent the scattering coefficient, $b_{scat}$. The

shaded areas represent the dust events periods.


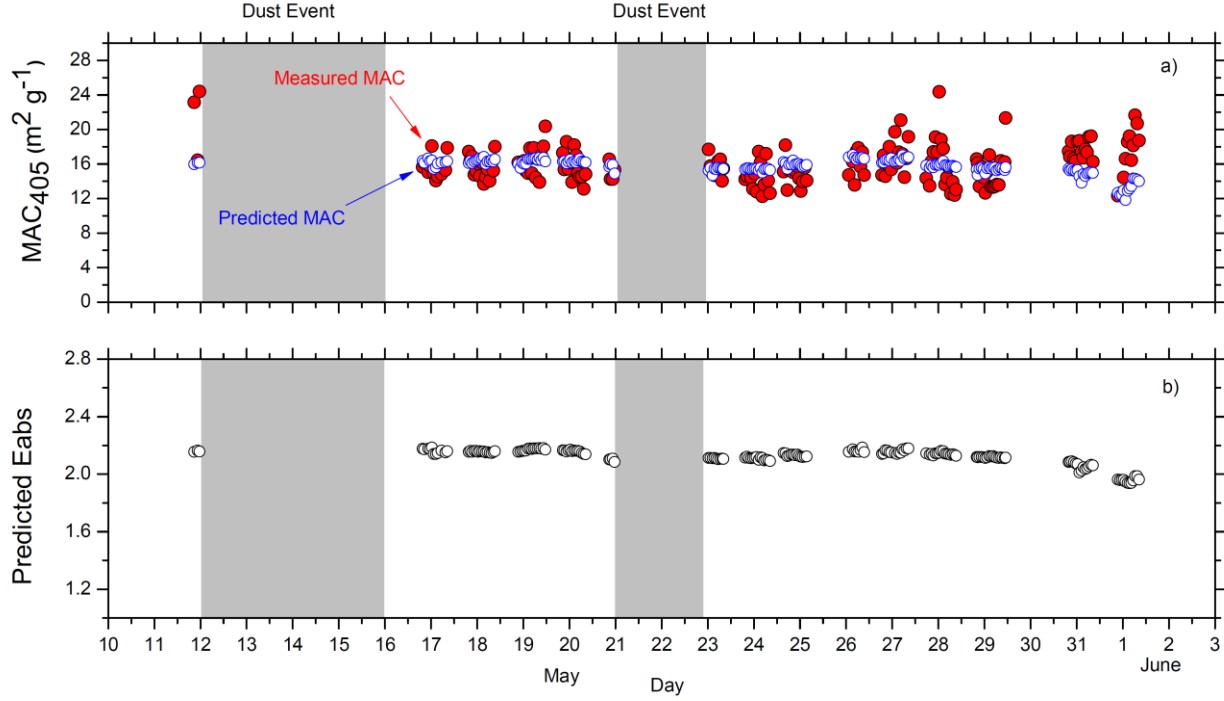


**Figure 6:** Hourly averaged results from the Mie theory calculations, assuming a non-absorbing

shell ($k$=0): a) predicted (blue circles) and measured MAC$_{405}$ (red circles). b) Hourly averaged

Predicted $E_{abs}$ values. The shaded areas represent the dust events periods.


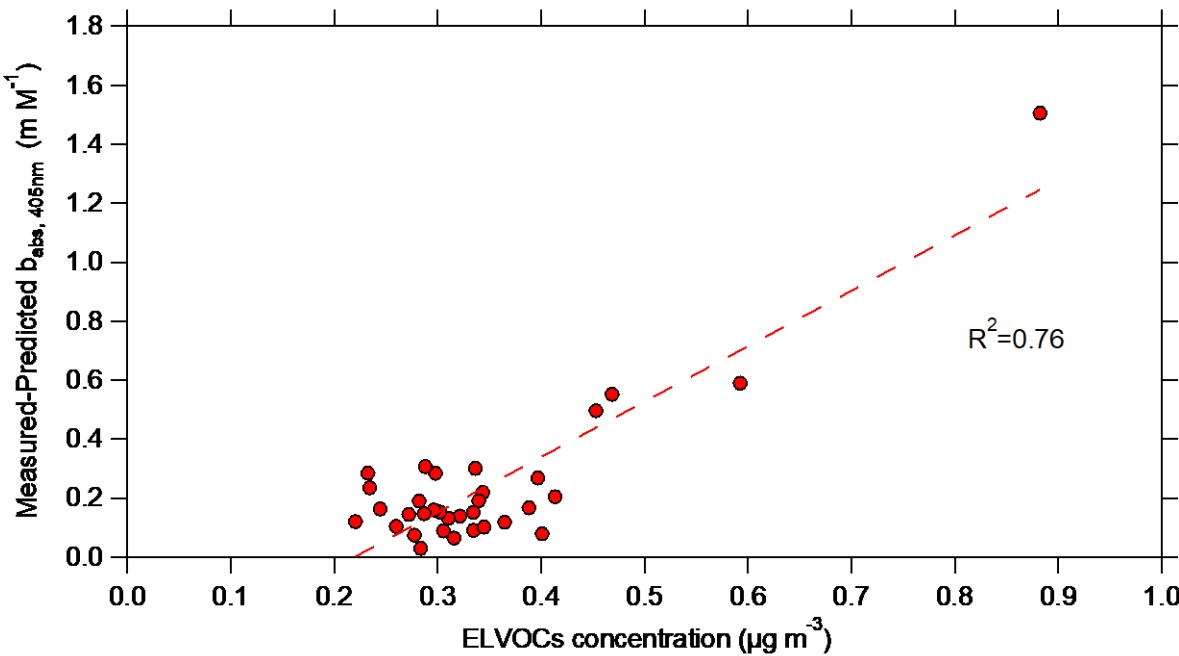


**Figure 7:** The difference between the measured and the predicted unexplained $b_{abs,405}$ as a function of the
estimated concentration of the ELVOCs. The data shown represent 3-hours averaged values.