# Peer review of "Aerosol light absorption and the role of extremely low volatility organic"

_Atmospheric Chemistry and Physics, 2019_

## Referee Comment (RC1) · Anonymous Referee #1 · 28 Jan 2020

The authors provide an analysis of measurements of light absorption, particle composition & volatility, and black carbon concentrations wherein they aim to establish the contribution of non-BC absorption to the total absorption and establish relationships with the observed particle composition. They conclude that the observed absorption is in great excess of that expected by uncoated black carbon alone and attribute this difference, based on comparison with Mie theory calculations, to a combination of the lensing effect for BC and to brown carbon. They conclude that that the lensing effect varies little over the campaign. The also conclude that the brown carbon absorptivity correlates with the absolute concentration of ELVOCs in their SOA. Unfortunately, I find their analysis and interpretation contains some fundamental flaws and misunderstandings regarding the relationship between Mie theory and observations. While I wish I

could be more positive, I do not think that this paper can be accepted in its current form and I question whether it would be acceptable at any point. My reasoning follows below.

1. The authors measure BC concentrations using an SP2. Not shown are any size distributions, which are important to consider as the measurements may have a negative bias owing to BC particles that are outside the detection window. This contribution may be small, but it should be considered if it has not been (it is not clear from the presentation). One way this has been dealt with in the literature is through single- or multi-modal fitting. This is another reason that the observed MAC might be higher than an expected value for pure BC (see line 278). This links to a question about measurement uncertainties, which are not reported. The authors must provide a discussion of uncertainties, that includes potential biases owing to factors such as BC outside of the measurement size window.

2. A major concern that I have about this paper relates to their interpretation of their absorption measurements. They state that the MAC of pure, uncoated BC at 405 nm should be 9.9 m2/g, but that they observe an average of 16.3 m2/g. However, they conclude later that their predicted value of 14.1 m2/g is a factor of two greater than it should be. This implies an MAC of 7.05 m2/g for pure BC from the calculations, much smaller than the 9.9 m2/g. This is because Mie theory generally leads to underestimates of the BC absorption. Thus, the calculated MAC is biased low. The extent of the underestimate depends on the details of the calculations (refractive index, particle size), which are not provided. The amount of information regarding the calculations is insufficient to allow clear judgement of their appropriateness. Regardless, it is evident that the authors are not making a fair comparison; the 14.1 m2/g value cannot be compared to the 16.3 m2/g value as they are starting from different reference values. Thus, the authors conclusion on L290 that there must be non-refractory absorbing material is not justified, nor are any subsequent calculations of the "delta_MAC". Related, the conclusions and calculations regarding the brown carbon imaginary RI, as determined

via closure, are not justified, in my opinion.

3. I do not find that the inputs to their calculations are sufficiently constrained to allow for accurate calculation of the absorption. Certainly, their calculated value cannot be directly compared to the observations (as the authors do) owing to my point in comment 3 that they undoubtedly underestimate the absorption by BC in their calculations. Beyond that, the authors seem to make a very poor assumption regarding the coatings on BC. The note that they use the ratio of the total aerosol mass divided by BC mass to estimate coating thickness. This is not appropriate. The BC-containing particles likely make up a small fraction of the total particles. Thus, some unknown fraction of the total non-BC material is internally mixed with BC. Quite often this fraction can be quite small, although it depends explicitly on the history of the air mass. If the authors are unable to provide constraints on the actual amount of coating on their BC, their calculated Eabs must be taken as an upper limit, with the actual value falling somewhere between 1 and the upper limit (2.07); unfortunately, no tighter constraint is possible without additional information regarding the true coating state. It is evident that the assumption that all material coats BC is a poor assumption, as the variability in the calculated MAC is negligible; this is simply because the particles always have "thick" coatings, in the plateau of the Mie curve, and thus little variability in the calculated MAC. The calculated MAC cannot be compared to the observations in any sort of quantitative manner. The conclusion that the lensing-induced enhancement is 2.07 is not justified. The authors might consider comparing the observed MAC_405 versus the NRPM to BC ratio; if the authors assumption that there is a substantial absorption enhancement and that all NRPM material is coated on BC were valid they should see a strong relationship between the MAC_405 and the NRPM/BC ratio that trends towards the expected pure BC value as the NRPM/BC ratio declines. (There may still be a relationship between the coating amount and the total NRPM/BC ratio, and thus even if a relationship is observed it is not definitive proof. Nonetheless, it might provide guidance for the interpretation.)

[Figure]

4. The authors report a strong correlation between the MAC at 405 nm and the BC concentration. They do not report the relationship at other wavelengths. If BrC is making a major contribution, and if it does not come from primary emissions that are associated with the BC emissions, then there should be a stronger relationship at longer wavelengths. The authors have measurements at longer wavelengths from the Aethelometer. Is the R^2 value larger at longer wavelengths than it is at shorter wavelengths?

5. One of the conclusions of Karnezi et al. (2014) is that "Our results indicate that existing TD-based approaches quite, often cannot estimate reliably the OA volatility distribution, leading to large uncertainties, since there are many different combinations of the three properties that can lead to similar thermograms." Yet, this is exactly what the authors have done here. It is thus unclear how the authors know that they have a unique solution, especially with respect to the co-variation between the derived volatility distribution and accommodation coefficient.

6. Why would one expect that an intensive property (the delta_MAC) should correlate with an extensive property (the ELVOC concentration)? Typically, intensive properties should correlate with some other intensive property (for example, the ELVOC fraction of total OA). Extensive properties should correlate with extensive properties (for example, the estimated unexplained absorption vs the absolute ELVOC concentration). I do not see a justification for why an intensive measurement should correlated with an extensive. The authors highlight this relationship in their abstract, yet spend a total of 7 lines presenting and discussing it. This is, in my opinion, insufficient. Does the delta_MAC correlate with any other measured properties, as but one thing that it would be useful for the authors to consider? That said, I'll note again that I do not believe the delta_MAC values are valid for the reasons discussed above, and thus, in my opinion, this entire analysis is suspect.

7. Also, what might be the source of these low-volatility absorbing organic components in this environment? This is not discussed. Why would the low-volatility components from LO-OOA have the same absorptivity as those from MO-OOA, as implied in the

equation at line 300. It is generally thought that OA from biogenic sources is relatively non-absorbing. Certainly, I am unaware of any measurements that indicate that SOA from biogenics, which presumably dominate this site, can have an RI as large as 0.4, which the authors indicate was measured. The authors need to provide justification via comparison to the literature.

8. The authors measured light absorption coefficients using an aethelometer and a PAX. As best I can tell, the authors have not compared the measurements from these two instruments to establish whether the aethelometer suffers from any positive biases that are known to impact filter-based absorption measurements. That said, such biases are less likely to influence the wavelength dependence measurements than they are the absolute absorption values.

Additional comment:

The authors must place their data in an appropriate repository.

---

## Referee Comment (RC2) · Anonymous Referee #2 · 10 Apr 2020

Review of Aerosol light absorption and the role of extremely low 1 volatility organic 2 compounds, by Tasoglou et al., for ACPD, 2020

Overview: The paper is focused on presenting observations from a ∼3 week measurement campaign at a Cretan site with little direct local influence. Measurements of rBC, aerosol light absorption, gas phase tracers, and bulk aerosol composition were complied to answer a primary focus topic: connecting observed aerosol absorption and potential BrC influences via the proxy of low volatility compounds.

Publication of this kind of work falls within the aims of ACP, and contributes to the broad understanding of aerosol absorption.

The manuscript does not yet make a strong enough case for its conclusions. The major

reasons for this are 1) the lack of uncertainty analysis 2) un-discussed conflict between conclusions and observations, and 3) lack of details provided about interpretation and work such that the reader is not sure what was done and what it means. In my opinion, the paper is a good start towards a publication ready manuscript, but more analysis work and explanation is still required to make a solid contribution to establishing distilled conclusions from the observations.

General Comments:

1) Uncertainty: throughout the manuscript consideration and discussion of uncertainty should be expanded. Comparison of MAC from theory and measurement depends critically on the uncertainties of the various approaches. To the extent that there is validating information about the PMF, this could also be considered in terms of uncertainty in drawing associations between bulk composition (which is all that the PMF is indicating) and the BC sources/microphysics (which also can affect MAC).

2) Some issues have not been sufficiently addressed. They include: a) how is the observation of AAE of ∼1 consistent with expectations for significant BrC absorption? In fact, can any significant absorption be attributed to low volatility spiecies? b) how is absorption of internal mixtures with rBC separated from bulk phase absorption? c) How is the PMF validated? Did rBC concentration (not deviation in MAC) correlate or anti-correlate with either factor? d) How are coating thicknesses constrained for rBC? e) How high is confidence, based on measurement and theoretical uncertainties, that the range of MAC is not solely associated with 1) rBC and PAS uncertainties or 2) lensing enhancements?

3) The reader needs more information about what was done in order to understand the results: a) How did the PAS and Aethalometer compare? Which was used to generate MAC405? How significant were the differences between them? b) SP2 - laser intensity? size range of detection for rBC? Correction to account for total rBC accumulation mode concentrations? Type (8 channel?). Precision of cal with CPMA?

Total uncertainty estimate in concentration (noting that different BC does have different response in SP2)? Calculation of coating thicknesses? c) Sampling: was the aerosol dried? What temperature was the laboratory? If it was not dried, the whole analysis is likely questionable. How large were the corrections for diffusion and thermophoresis losses in the denuder (I expect likely negligible for Accum. mass). d) Mie theory tests: what constraints on coating thickness were used in the Mie Theory evaluation of index of refraction? Was this derived from SP2 data? How? This is a critical question if there is to be any conclusion drawn from this analysis. Note that SP2 determination of coating thicknesses are highly uncertain for thin coatings (e.g. Ohata et al.,(2016), Hygroscopicity of materials internally mixed with black carbon measured in Tokyo, J. Geophys. Res. Atmos., 121, doi:10.1002/2015JD024153.).

Line-level and specific comments L37 - averaged over a population. The literature review 34-83 - lensing enhancement is well written, but might give the reader the incorrect idea that similar work was done here. As only the AMS and SMPS were behind denuder this is not the case.

(Note that it is a nice feature of your data set that initial brown carbon bleaching/evap should have already substantially occurred )

TD - please provide more info about the temperature set points so that the reader isn't surprised by the lack of data between 200 and $\sim$375 °C.

L197 Bond and Bergstrom, 2006 suggest (1.95,-0.79) for the complex index at 1.8 g/cc. Not as stated.

The paper by China et al., 10.1002/2014GL062404 will be useful for justifying use of Mie theory shell and core.

Please consider if Figure 2 is unnecessary given figure 4Ăă? Combine?

Wavelength of Mie theory calcs? Note that RI for BC is likely low. Were the calculations carried out for observed core size dist? coating thickness? On what time scale were

[Figure]

those calculated?

Figure 7: negative Measure-Predicted not included - meaningless without all the data. It appears that some positive M-P data is also missing (based on the number of data points in figure 7 vs those in figure 6a).

L249: "average calculated saturation concentration of 0.016 ug/m3"

Time series not very helpful for understanding more than trends in extensive properties (amounts). Scatter plots would be more useful.

L264: "similarly" - actually Babs and Bscat appear poorly correlated.

L273 - please include the uncertainty on this value. L286: Not clear how coating thicknesses were constrained. Big issue./ L281: AAE only described for 523 to 405… how the range calculated, averaged?

L289: is 13% meaningful in the context of uncertainties? L313 - I think you mean "no periods of enhanced BB influence"

––––––––––––––––––––––––––––––––

---

## Author Comment (AC1) · 21 May 2020

**(1)** *The authors measure BC concentrations using an SP2. Not shown are any size distributions, which are important to consider as the measurements may have a negative bias owing to BC particles that are outside the detection window. This contribution may be small, but it should be considered if it has not been (it is not clear from the presentation). One way this has been dealt with in the literature is through single- or multi-modal fitting. This is another reason that the observed MAC might be higher than an expected value for pure BC (see line 278). This links to a question about measurement uncertainties, which are not reported. The authors must provide a discussion of uncertainties, that includes potential biases owing to factors such as BC outside of the measurement size window.*

The BC number concentration distributions were fitted in our baseline calculations using a Gaussian distribution to account for particles smaller than the SP2 detection limit (Ditas et al., 2018). This extrapolation resulted in an increase of the BC mass concentration by 3-8 percent, so its effect on the reported results (e.g., the observed MAC) was minor. This information has been added to the revised paper together with a graph (in the Supplementary Information) with a few representative mass distributions. A discussion of other SP2 uncertainties has been also added (Lack et al., 2012). This SP2 issue is discussed together with the rest of the uncertainties of both the measurements and the modeling in a new uncertainty analysis section that has been added to the revised paper.

**(2)** *A major concern that I have about this paper relates to their interpretation of their absorption measurements. They state that the MAC of pure, uncoated BC at 405 nm should be 9.9 m²/g, but that they observe an average of 16.3 m²/g. However, they conclude later that their predicted value of 14.1 m²/g is a factor of two greater than it should be. This implies a MAC of 7.05 m²/g for pure BC from the calculations, much smaller than the 9.9 m²/g. This is because Mie theory generally leads to underestimates of the BC absorption. Thus, the calculated MAC is biased low. The extent of the underestimate depends on the details of the calculations (refractive index, particle size), which are not provided. The amount of information regarding the calculations is insufficient to allow clear judgement of their appropriateness. Regardless, it is evident that the authors are not making a fair comparison; the 14.1 m²/g value cannot be compared to the 16.3 m²/g value as they are starting from different reference values. Thus, the authors conclusion on L290 that there must be non-refractory absorbing material is not justified, nor are any subsequent calculations of the "deltaMAC". Related, the conclusions and calculations regarding the brown carbon imaginary RI, as determined.*

This is a valid point but it is due to a misunderstanding of this rather poorly placed sentence. Our calculations are self-consistent. The MAC of 9.9 m²/g was never used in any of the calculations and should not be part of the discussion here. It was mentioned

at this point just to support the argument that the measured value of 16.3 m$^2$/g is too high compared to what would be expected for pure uncoated BC. We have moved the discussion of the MAC values of pure, uncoated BC to the introduction of the paper stressing that its value at a specific wavelength depends both on the size distribution of the particles but also on their degree of aging (fresh aggregates versus collapsed more spherical structures). We just make the point now that the measured value of 16.3 m$^2$/g is far outside the range expected for pure, uncoated BC particles.

The details of all the Mie theory calculations are now provided at one place in the paper. Some of them were in different parts of the original work. The predicted average BC MAC405 using the measured BC size distribution and assuming pure uncoated spherical particles was 7.2 m$^2$/g varying from 6.1 to 7.8 m$^2$/g during the study (hourly averages). This is consistent with the expected 5.3-9.2 m$^2$/ g for BC core diameters in the 10-350 nm size range. This information has been also added to the paper.

We should also stress that the 16.3 m$^2$/g is the average measured value based on the PAX and the SP2 measurements. It is not based on an assumed reference value for the pure, uncoated BC MAC value. Thus, there is no issue regarding the consistency of the predicted and the measured values. The uncertainty of both is analyzed in the new uncertainty analysis section. Please note that the differences of the measured and the predicted MAC405 were as high as 7 m$^2$/g and we argue that this difference cannot be explained by the corresponding uncertainties in the measurements and the modeling.

**(3)** *I do not find that the inputs to their calculations are sufficiently constrained to allow for accurate calculation of the absorption. Certainly, their calculated value cannot be directly compared to the observations (as the authors do) owing to my point in comment 2 that they undoubtedly underestimate the absorption by BC in their calculations. Beyond that, the authors seem to make a very poor assumption regarding the coatings on BC. The note that they use the ratio of the total aerosol mass divided by BC mass to*

*estimate coating thickness. This is not appropriate. The BC-containing particles likely make up a small fraction of the total particles. Thus, some unknown fraction of the total non-BC material is internally mixed with BC. Quite often this fraction can be quite small, although it depends explicitly on the history of the air mass. If the authors are unable to provide constraints on the actual amount of coating on their BC, their calculated Eabs must be taken as an upper limit, with the actual value falling somewhere between 1 and the upper limit (2.07); unfortunately, no tighter constraint is possible without additional information regarding the true coating state. It is evident that the assumption that all material coats BC is a poor assumption, as the variability in the calculated MAC is negligible; this is simply because the particles always have "thick" coatings, in the plateau of the Mie curve, and thus little variability in the calculated MAC. The calculated MAC cannot be compared to the observations in any sort of quantitative manner. The conclusion that the lensing-induced enhancement is 2.07 is not justified. The authors might consider comparing the observed MAC405 versus the NRPM to BC ratio; if the authors assumption that there is a substantial absorption enhancement and that all NRPM material is coated on BC were valid they should see a strong relationship between the MAC405 and the NRPM/BC ratio that trends towards the expected pure BC value as the NRPM/BC ratio declines. (There may still be a relationship between the coating amount and the total NRPM/BC ratio, and thus even if a relationship is observed it is not definitive proof. Nonetheless, it might provide guidance for the interpretation.)*

The uncertainty of the estimated value of the absorption of the BC is addressed in the new uncertainty analysis section. We test the hypothesis that this absorption has been underestimated and that this can explain the difference in measured and predicted absorption of the particles. We show that while increasing the absorption of the BC cores can explain the average observed absorption it cannot explain its variability.

The second point of the reviewer is that our calculations are based on the assumption that all particles of the same size contain the same BC (internally mixed aerosol population). This is correct and it does provide the upper limit for the lensing effect and

therefore the minimum of the unexplained absorption that could be assigned to BrC. All the air masses sampled during the campaign were quite aged so it is reasonable to assume that all the BC particles were coated. In order to provide a better constraint for our method we calculated the coating thickness of the rBC cores using the leading-edge-only (LEO) fit method (Gao et al. 2007) and the SP2 data. The thicknesses as expected were lower than those resulting from the internal mixing assumption but were still significant. We have repeated our calculations using this estimate of the coating thickness and tested the robustness of the link between the unexplained absorption and the ELVOC concentrations but also for the rest of our results.

We have followed the reviewer's suggestion and investigated the relationship between MAC405 and the ratio of non-refractory PM1 and rBC. There is a positive correlation between the two (R=0.31) and the intercept is 12.8 m$^2$/g. This analysis provides, as the reviewer suggests, limited support to our argument, but has been added to the paper nonetheless.

**(4)** *The authors report a strong correlation between the MAC at 405 nm and the BC concentration. They do not report the relationship at other wavelengths. If BrC is making a major contribution, and if it does not come from primary emissions that are associated with the BC emissions, then there should be a stronger relationship at longer wavelengths. The authors have measurements at longer wavelengths from the Aethalometer. Is the $R^2$ value larger at longer wavelengths than it is at shorter wavelengths?*

We followed the reviewer's suggestion and examined the values of the $R^2$ between the MAC and the rBC concentration at different wavelengths. Indeed, the $R^2$ at 370 nm was 0.67, a value lower than the $R^2$ at 405 nm (equal to 0.74). This is consistent with the presence of some BrC. However, the first is based on the Aethalometer and the second on the PAX measurements. Moving at higher wavelengths the $R^2$ at 450 nm was 0.68, at 520 nm 0.67, at 590 nm 0.65, and at 660 nm 0.65. These values provide

a mixed picture. Part of the explanation for this behavior could be that a fraction of the BrC is associated with rBC either from the emissions or from the associated gas-phase pollutants that react in their way to the site. This information has been added to the revised paper and the corresponding figures to the Supplementary Information.

**(5)** *One of the conclusions of Karnezi et al. (2014) is that "Our results indicate that existing TD-based approaches quite, often cannot estimate reliably the OA volatility distribution, leading to large uncertainties, since there are many different combinations of the three properties that can lead to similar thermograms." Yet, this is exactly what the authors have done here. It is thus unclear how the authors know that they have a unique solution, especially with respect to the co-variation between the derived volatility distribution and accommodation coefficient.*

This is a valid point. The comment of Karnezi et al. (2014) referred to a detailed volatility distribution in the Volatility-Basis-Set framework. We have recently shown (Louvaris et al., 2017; Cain et al., 2020) that the uncertainty of volatility classes (e.g. ELVOCs, LVOCs, SVOCs) is significantly less than the uncertainty of individual volatility bins. The uncertainty of these concentrations is provided by the fitting algorithm used in the paper and is shown in Figure 4 of the paper. We have repeated the analysis of the links between the unexplained absorption and the ELVOC levels. The results are discussed in the new uncertainty analysis section and show that our original conclusion is robust.

**(6)** *Why would one expect that an intensive property (the deltaMAC) should correlate with an extensive property (the ELVOC concentration)? Typically, intensive properties should correlate with some other intensive property (for example, the ELVOC fraction of total OA). Extensive properties should correlate with extensive properties (for example, the estimated unexplained absorption vs the absolute ELVOC concentration). I do not see a justification for why an intensive measurement should correlated with an*

*extensive. The authors highlight this relationship in their abstract, yet spend a total of 7 lines presenting and discussing it. This is, in my opinion, insufficient. Does the deltaMAC correlate with any other measured properties, as but one thing that it would be useful for the authors to consider? That said, I'll note again that I do not believe the deltaMAC values are valid for the reasons discussed above, and thus, in my opinion, this entire analysis is suspect.*

This is a valid point and we agree with the reviewer that it is preferable to correlate either intensive properties or extensive properties to each other. We have examined the correlation of the unexplained babs and the ELVOC concentration and found an $R^2=0.76$. This actually strengthens our argument regarding the contribution of the ELVOCs to the unexplained absorption by removing the other factors entering into the calculation of the intensive property the MAC. We have revised Figure 7 accordingly.

Please note that we devote two pages in the original manuscript explaining the estimation of the ELVOC concentrations and a few more pages discussing the calculation of the unexplained absorption.

We have examined the correlations of $\Delta$MAC with the rBC, with other parameters and found that they were all much lower than those for the ELVOCs (0.76). For example the $R^2$ with the rBC was 0.38 (rBC), with the LO-OOA 0.44 and with the MO-OOA 0.29. There was a relatively high $R^2$ with the sulfate levels (0.59) that could be interesting as high sulfate levels in this area correspond to high aerosol acidity which has been shown to promote formation of oligomers in secondary organic aerosol. This analysis has been added to the paper.

**(7)** *Also, what might be the source of these low-volatility absorbing organic components in this environment? This is not discussed. Why would the low-volatility components from LO-OOA have the same absorptivity as those from MO-OOA, as implied in the equation at line 300. It is generally thought that OA from biogenic sources is relatively non-absorbing. Certainly, I am unaware of any measurements that indicate that SOA*

*from biogenics, which presumably dominate this site, can have an RI as large as 0.4, which the authors indicate was measured. The authors need to provide justification via comparison to the literature.*

The OA in the site is the result of long range transport from other areas and is not due to local sources (including biogenics). During the period of the study the area near the station has almost no vegetation (mainly dried out bushes). The average O:C of the OA in the site (around 0.83 on average) during the campaign is a lot higher than that of fresh biogenic SOA. Air mass came to the site from continental Greece and Balkans, Turkey, Africa and central Mediterranean. The OA in the Eastern Mediterranean during the summer has significant contributions from both anthropogenic and biogenic sources but their contributions remain uncertain. Based on our recent work (Drosatou et al., 2019) the LO-OOA and MO-OOA do not reflect different sources, but rather different degrees of chemical aging. ELVOCs can be produced both by primary sources (combustion of fossil fuels but also biomass burning) and secondary processes. Given the intense chemical processing of the organic compounds from all sources in their way to Finokalia, we cannot assign based on our measurements to a specific source or sources. The association of BrC with material of lower volatility away from its sources has been reported in a number of studies focusing on biomass burning (Saleh et al., 2014; Wong et al., 2019). This discussion has been added to the revised paper.

**(8)** *The authors measured light absorption coefficients using an aethalometer and a PAX. As best I can tell, the authors have not compared the measurements from these two instruments to establish whether the aethalometer suffers from any positive biases that are known to impact filter-based absorption measurements. That said, such biases are less likely to influence the wavelength dependence measurements than they are the absolute absorption values.*

A comparison of the aethalometer and the PAX measurements has been added to the Supplementary Information of the revised paper. We compared the absorption coefficients measured by the aethalometer at 370 nm and at 450 nm with that measured by the PAX at $\lambda$=405 nm. The measurements of the two instruments were highly correlated with $R^2$=0.9-0.91. The babs,370 measured by the aethalometer was higher than the babs,405 of the PAX. Their relationship is described by the equation y=3.32 x - 3.24. Similarly, the babs,450 measured by the Aethalometer was higher than that of the PAX at 405 nm and their relationship was described by the equation y=1.85 x – 0.93. These discrepancies could possibly exist due to the artifacts associated with OA loadings on the filter (Cappa et al., 2008; Lack et al., 2008). A brief discussion has been added in the main paper and the corresponding figures have been added to the SI.

**References**

Cappa, C., Lack, D., Burkholder, J., and Ravishankara, A.: Bias in filter based aerosol light absorption measurements due to organic aerosol loading: Evidence from laboratory measurements, Aerosol Sci. Technol., 42, 1022–1032, 2008.

Cain, K. P., Karnezi, E., Pandis, S. N.: Challenges in determining atmospheric organic aerosol volatility distributions using thermal evaporation techniques, Aerosol Sci. Tech., doi: 10.1080/02786826.2020.1748172, 2020.

Ditas, J., Ma, N., Zhang, Y., Assmann, D., Neumaier, M., Riede, H., Karu, E., Williams, J., Scharffe, D., Wang, Q., Saturno, J., Schwarz, J. P., Katich, J. M., McMeeking, G. R., Zahn, A., Hermann, M., Brenninkmeijer, C. A. M., Andreae, M. O., Pöschl, U., Su, H., and Cheng, Y.: Strong impact of wildfires on the abundance and aging of black carbon in the lowermost stratosphere, P. Natl. Acad. Sci. USA, 115, 11595–11603, 2018.

Drosatou, A. D., Skyllakou, K., Theodoritsi, G. N., and Pandis, S. N.: Positive matrix factorization of organic aerosol: insights from a chemical transport model, Atmos. Chem. Phys., 19, 973–986, https://doi.org/10.5194/acp-19-973-2019, 2019.

Gao, R. S., Schwarz, J. P., Kelly, K. K., Fahey, D. W., Watts, L. A., Thompson, T. L.,

Spackman, J. R., Slowik, J. G., Cross, E. S., Han, J. H., Davidovits, P., Onasch, T. B., and Worsnop, D. R.: A novel method for estimating light-scattering properties of soot aerosols using a modified single-particle soot photometer, Aerosol Sci. Technol., 41, 125–135, 2007.

Karnezi, E., Riipinen, I. and Pandis, S. N.: Measuring the atmospheric organic aerosol volatility distribution: A theoretical analysis, Atmos. Meas. Tech., 7, 2953–2965, 2014.

Lack, D. A., J. M. Langridge, R. Bahreini, C. D. Cappa, A. N. Middlebrook, and J. P. Schwarz.: Brown carbon and internal mixing in biomass burning particles. Proc. Natl. Acad. Sci. USA, 109, 14802–14807, 2012.

Lack, D. A., Cappa, C. D., Covert, D. S., Baynard, T., Massoli, P., Sierau, B., Bates, T. S., Quinn, P. K., Lovejoy, E. R., and Ravishankara, A. R.:Bias in filter based aerosol light absorption measurements due to organic aerosol loading: Evidence from ambient measurements, Aerosol Sci. Technol., 42, 1033–1041, 2008.

Louvaris, E. E., Florou, K., Karnezi, E., Papanastasiou, D. K., Gkatzelis, G. I., and Pandis, S. N.: Volatility of source apportioned wintertime organic aerosol in the city of Athens, Atmos. Environ.,158, 138-147, 2017.

Saleh, R., Robinson, E. S., Tkacik, D. S., Ahern, A. T., Liu, S., Aiken, A. C., Sullivan, R. C., Presto, A. A., Dubey, M. K., Yokelson, R. J., Donahue, N. M., and Robinson, A. L.: Brownness of organics in aerosols from biomass burning linked to their black carbon content, Nat. Geosci., 7, 647–650, 2014.

Wong, J. P. S., Tsagkaraki, M., Tsiodra, I., Mihalopoulos, N., Violaki, K., Kanakidou, M., Sciare, J., Nenes, A., and Weber, R. J.: Atmospheric evolution of molecular-weight-separated brown carbon from biomass burning, Atmos. Chem. Phys., 19, 7319–7334, https://doi.org/10.5194/acp-19-7319-2019, 2019.
* * *

---

## Author Comment (AC2) · 21 May 2020

**(1)** *Overview: The paper is focused on presenting observations from a âĹij3 week measurement campaign at a Cretan site with little direct local influence. Measurements of rBC, aerosol light absorption, gas phase tracers, and bulk aerosol composition were complied to answer a primary focus topic: connecting observed aerosol absorption and potential BrC influences via the proxy of low volatility compounds. Publication of this kind of work falls within the aims of ACP, and contributes to the broad understanding of aerosol absorption. The manuscript does not yet make a strong enough case for its conclusions. The major reasons for this are 1) the lack of uncertainty analysis 2) un-discussed conflict between conclusions and observations, and 3) lack of details provided about interpretation and work such that the reader is not sure what was done*

[Figure]

*and what it means. In my opinion, the paper is a good start towards a publication ready manuscript, but more analysis work and explanation is still required to make a solid contribution to establishing distilled conclusions from the observations.*

We appreciate the constructive comments and suggestions of the reviewer. To address them we have: (1) performed additional uncertainty analysis of the results focusing on the robustness of our conclusions. A new uncertainty analysis section has been added; (2) addressed the apparent conflict between the value of the angstrom exponent and the conclusion about the brown carbon levels; and (3) provided additional details about the approach used. Our detailed responses (in regular font) follow each reviewer's comment (in italics).

*General Comments*

**(2)** *Uncertainty: throughout the manuscript consideration and discussion of uncertainty should be expanded. Comparison of MAC from theory and measurement depends critically on the uncertainties of the various approaches. To the extent that there is validating information about the PMF, this could also be considered in terms of uncertainty in drawing associations between bulk composition (which is all that the PMF is indicating) and the BC sources/microphysics (which also can affect MAC).*

We have added a new section in the paper describing our uncertainty analysis of the results. This takes into account both the uncertainty of the measurements but also the uncertainty introduced by the various assumptions during the theoretical analysis. We have followed the reviewer's suggestion and addressed the uncertainty of the PMF analysis. We used the bootstrapping approach (Ulbrich et al., 2009) performing ten additional simulations. The estimated uncertainty for the LO-OOA concentrations was 2 percent and for the MO-OOA concentrations was 3 percent. These relatively small uncertainties suggest that the PMF uncertainty regarding the determination of these factors does not affect our conclusions. The corresponding results have been added to the Supplementary Information and are discussed in the new uncertainty analysis sec-

tion. We have also looked at the correlations between the concentrations of MO-OOA, LO-OOA and BC. The $R^2$ between the hourly concentrations of MO-OOA and rBC was 0.12, and for LO-OOA and rBC 0.17. This is not unexpected given that Finokalia is far away from the corresponding sources of both BC and organic compounds and significant physical and chemical processing has taken place during the transport of the aerosol from the sources to the receptor. The $R^2$ between the hourly concentrations of the ELVOCs and rBC was 0.25. This information has been added to the revised paper.

**(3)** *Some issues have not been sufficiently addressed. They include: a) how is the observation of AAE of around 1 consistent with expectations for significant BrC absorption? In fact, can any significant absorption be attributed to low volatility species?*

The observation of AAE around 1 is not inconsistent with the presence of some BrC. For example, Lack and Cappa (2010) argued that an AAE<1.6 does not exclude the possibility of BrC; rather BrC cannot be confidently assigned unless AAE>1.6. The AAE of coated BC is highly sensitive to particle size distribution and it decreases as particle size increases (Liu et al., 2018). This last study shows that relatively low values of AAE should be expected in a site like Finokalia and demonstrated the importance of various parameters on the BC AAE and the potential problems introduced by assuming BC AAE as being equal to 1.0. Finally, please note that based on our estimates BrC led to a 15 average increase of the campaign average babs at 405 nm. This increase is on average modest and is not inconsistent with the relatively low AAE observed in this remote site with relatively large, heavily processed particles. The finding that this unexplained absorption increases as the concentration of the ELVOCs increases is probably the most interesting finding of our work. We have added discussion of these points in the revised paper.

**(4)** *b) how is absorption of internal mixtures with rBC separated from bulk phase absorption?*

The assumption in our baseline analysis is that the non-refractory absorbing material is coating the BC particles. All the air masses sampled during the campaign were quite aged so it is reasonable to assume that all the BC particles were coated. Our baseline calculations are based on the assumption that all particles of the same size contain the same BC (internally mixed aerosol population). This provides the upper limit for the lensing effect and therefore the minimum of the unexplained absorption that could be assigned to BrC. We have performed an additional calculation using the coating estimated by the SP2 measurements to test the robustness of our conclusions. Additional assumptions are examined in the added uncertainty analysis section.

**(5)** *c) How is the PMF validated? Did rBC concentration (not deviation in MAC) corre-late or anti-correlate with either factor?*

We used the bootstrapping approach (Ulbrich et al., 2009) performing ten additional simulations. The estimated uncertainty for the LO-OOA concentrations was 2 percent and for the MO-OOA concentrations was 3 percent. These relatively small uncertainties suggest that the PMF uncertainty regarding the determination of these factors does not affect our conclusions. Both MO-OOA and LO-OOA were positively correlated with rBC with the corresponding correlation coefficient R being 0.35 for MO-OOA and 0.41 for LO-OOA. These correlations between the absolute concentrations of the various PM components are in general expected in remote sites that are affected sometimes by more polluted air masses (leading to increases of the major PM components) and cleaner air masses processed by rain (leading to decreases of the major PM components. For example, the R of the non-refractory PM1 and the rBC was 0.55. The corresponding results have been added to the Supplementary Information and are discussed in the new uncertainty analysis section.

**(6)** *d) How are coating thicknesses constrained for rBC?*

Two different approaches are now used for the calculation of the coating thicknesses.

In the first approach used in the original paper the coating of the BC core was calculated using the PM1 mass and the rBC distribution as it was measured by the SP2. This approach corresponds to the internal mixing assumption. In order to provide constraints in our method we also calculated in the revised paper the coating thickness of the rBC cores using the leading-edge-only (LEO) fit method (Gao et al. 2007) of the SP2 data. The thicknesses as expected were lower than those resulting from the internal mixing assumption, but were still significant. We have repeated our calculations using this estimate of the coating thickness and tested the robustness of the link between the unexplained absorption and the ELVOC concentrations but also for the rest of our results.

**(7)** *e) How high is confidence, based on measurement and theoretical uncertainties, that the range of MAC is not solely associated with 1) rBC and PAS uncertainties or 2) lensing enhancements?*

A detailed uncertainty analysis has been added to the revised paper to address these issues. The PAX has an uncertainty of less than 10 percent for the aerosol absorption measurement (Lack et al. 2012). Part of the uncertainty of the rBC mass by the SP2 is related to the detection limit of the smaller BC particles. The BC distributions were fitted in our baseline calculations using a Gaussian distribution to account for particles smaller than the SP2 detection limit (Ditas et al., 2018). This extrapolation resulted in an increase of the BC mass concentration by 3-8 percent, so its effect on the reported results (e.g., the observed MAC) was minor. The average propagated uncertainty of the MAC405, based on the measurements of rBC and babs at $\lambda$=405 nm is approximately 20 percent. Mie theory calculation can underestimate the absorption due to BC morphology and mixing. These uncertainties are discussed together with the rest of the uncertainties of the measurements and the modeling in a new uncertainty analysis section that has been added to the revised paper.

**(8)** *The reader needs more information about what was done in order to understand the*

*results: a) How did the PAS and Aethalometer compare? Which was used to generate MAC405? How significant were the differences between them?*

A comparison of the aethalometer and the PAX measurements has been added to the Supplementary Information of the revised paper. We compared the absorption coefficients measured by the aethalometer at 370 nm and at 450 nm with that measured by the PAX at $\lambda$=405 nm. The measurements of the two instruments were highly correlated with $R^2$=0.9-0.91. The babs,370 measured by the aethalometer was higher than the babs,405 of the PAX. Their relationship is described by the equation y=3.32 x - 3.24. Similarly, the babs,450 measured by the Aethalometer was higher than that of the PAX at 405 nm and their relationship was described by the equation y=1.85x – 0.93. These discrepancies could possibly exist due to the artifacts associated with OA loadings on the filter (Cappa et al., 2008; Lack et al., 2008). A brief discussion has been added in the main paper and the corresponding figures have been added to the SI. The MAC405 was calculated by dividing the babs,405 measured by the PAX and the rBC mass. This point is clarified in the revised manuscript to avoid confusion.

**(9)** *b) SP2 - laser intensity? size range of detection for rBC? Correction to account for total rBC accumulation mode concentrations? Type (8 channel?). Precision of cal with CPMA? Total uncertainty estimate in concentration (noting that different BC does have different response in SP2)? Calculation of coating thicknesses?*

The SP2 uses a Nd:YAG Laser at 1064 nm operated at 4 V and 3400 A. The calibration of the SP2 was verified in separate experiments using a centrifugal particle mass analyzer (CPMA, Cambustion). The SP2 mode mass and the CPMA mode mass were in good agreement, $R^2$=0.99 (Saliba et al., 2016). This information has been added to the experimental section.

The BC number concentration distributions were fitted in our baseline calculations using a Gaussian distribution to account for particles smaller than the SP2 detection limit (Ditas et al., 2018). This extrapolation resulted in an increase of the BC mass concentration by 3-8 percent, so its effect on the reported results (e.g., the observed MAC) was minor. This information has been added to the revised paper together with a graph (in the Supplementary Information) with a few representative mass distributions. A discussion of other SP2 uncertainties has been also added (Lack et al. 2012). This SP2 issue is discussed together with the rest of the uncertainties of both the measurements and the modeling in a new uncertainty analysis section that has been added to the revised paper.

Two different approaches are now used for the calculation of the coating thicknesses. In the first approach used in the original paper the coating of the BC core was calculated using the PM1 mass and the rBC distribution as it was measured by the SP2. This approach corresponds to the internal mixing assumption. In order to provide constraints in our method we also calculated in the revised paper the coating thickness of the rBC cores using the leading-edge-only (LEO) fit method (Gao et al. 2007) of the SP2 data. The thicknesses as expected were lower than those resulting from the internal mixing assumption, but were still significant. We have repeated our calculations using this estimate of the coating thickness and tested the robustness of the link between the unexplained absorption and the ELVOC concentrations but also for the rest of our results.

**(10)** *c) Sampling: was the aerosol dried? What temperature was the laboratory? If it was not dried, the whole analysis is likely questionable. How large were the corrections for diffusion and thermophoresis losses in the denuder (I expect likely negligible for Accum. mass).*

A diffusion drier was used upstream of the optical measurements to reduce any relative humidity-related measurement artifacts in the measurements (Arnott et al. 2003). The campaign average temperature and relative humidity were 22±4 C and 53±19 percent, respectively. The measurement station had an AC unit maintaining the temperature at 25 C.

To account for losses in the thermodenuder, sample flow rate as well as size- and temperature-dependent loss corrections were applied following Louvaris et al. (2017) corresponding to the operating conditions during the campaign. Up to 20 percent of the particulate matter was lost in the TD, at temperatures up to 100 C, while the losses increased for higher temperatures. For a temperature equal to 400 C almost 50 percent of particles larger than 50 nm is lost. The uncertainty introduced by the correction ranged was approximately 20 percent (Gkatzelis et al., 2016; Louvaris et al., 2017).

The above points have been added to the manuscript.

**(11)** *Mie theory tests: what constraints on coating thickness were used in the Mie Theory evaluation of index of refraction? Was this derived from SP2 data? How? This is a critical question if there is to be any conclusion drawn from this analysis. Note that SP2 determination of coating thicknesses are highly uncertain for thin coatings (e.g. Ohata et al. (2016). Hygroscopicity of materials internally mixed with black carbon measured in Tokyo, J. Geophys. Res. Atmos., 121, doi:10.1002/2015JD024153.)*

Two different approaches are now used for the calculation of the coating thicknesses. In the first approach used in the original paper the coating of the BC core was calculated using the PM1 mass and the rBC distribution as it was measured by the SP2. This approach corresponds to the internal mixing assumption. In order to provide constraints in our method we also calculated in the revised paper the coating thickness of the rBC cores using the leading-edge-only (LEO) fit method (Gao et al. 2007) of the SP2 data. The thicknesses as expected were lower than those resulting from the internal mixing assumption, but were still significant. We have repeated our calculations using this estimate of the coating thickness and tested the robustness of the link between the unexplained absorption and the ELVOC concentrations but also for the rest of our results. A coating refractive index of 1.55 was assumed following previous studies (Bond and Bengstrom, 2006).

**(12)** *Line-level and specific comments L37 - averaged over a population. The literature review 34-83 - lensing enhancement is well written, but might give the reader the incorrect idea that similar work was done here. As only the AMS and SMPS were behind denuder this is not the case. (Note that it is a nice feature of your data set that initial brown carbon bleaching/evap should have already substantially occurred).*

In the literature review we wanted to summarize the finding of previous studies but also the different techniques used. We clarify that in our study only the AMS and the SMPS were behind the thermodenuder and that an interesting feature of the dataset is that the organic aerosol has been thoroughly processed.

**(13)** *TD - please provide more info about the temperature set points so that the reader isn't surprised by the lack of data between 200 and 375 âŮęC.*

We have added the detailed information about the temperature steps used in the thermodenuder explaining the change from 200 to approximately 375 C.

**(14)** *L197 Bond and Bergstrom, 2006 suggest (1.95,-0.79) for the complex index at 1.8 g/cc. Not as stated.*

For the BC core we assumed a refractive index of the core nrBC=1.85+0.71i based on Bond et al. (2006). We have corrected the reference in the revised manuscript.

**(15)** *The paper by China et al., 10.1002/2014GL062404 will be useful for justifying use of Mie theory shell and core.*

We have added the suggested reference to the revised paper to further support the use of core and shell Mie theory for our site.

**(16)** *Please consider if Figure 2 is unnecessary given Figure 4a? Combine?*

We believe that the existence of Figure 2 is helping the reader for easy comparison between our study and previous studies in area (Lee et al., 2010). In addition, the plot is needed for comparison with other studies that use similar methods to calculate the OA volatility. Finally, the use of both Figure 2 and Figure 4 show how much each factor MO-OOA, and LO-OOA affects the total OA volatility.

**(17)** *Wavelength of Mie theory calcs? Note that RI for BC is likely low. Were the calculations carried out for observed core size dist? coating thickness? On what time scale were those calculated?*

The Mie theory calculation was performed at a wavelength of 405 nm in order to be able to compare our results with the measurements from the PAX. The sensitivity to the refractive index of BC was investigated and the results are discussed in the added uncertainty analysis section. The calculations were carried out based on the measured rBC core size distribution extrapolated so smaller sizes. Two different coating thicknesses were used. One based on the internal mixture assumption providing an upper limit for the calculation and one based on the thicknesses calculated by the SP2. One-hour averaged data were used as inputs in the model. The above information has been added to the revised paper.

**(18)** *Figure 7: negative Measure-Predicted not included - meaningless without all the data. It appears that some positive M-P data is also missing (based on the number of data points in figure 7 vs those in Figure 6a).*

Please note that the data in Figure 6a are one-hour averages while those in Figure 7 are 3-hour averages. This is now explained in the corresponding figure captions. The original Figure 7 has been redrawn using the unexplained absorption in the y-axis. We are now discussing the effect of the few negative unexplained values in the corresponding section and perform the corresponding statistical analysis both with and without them.

**(19)** *L249: "average calculated saturation concentration of 0.016 ug/m3".*

Corrected.

**(20)** *Time series not very helpful for understanding more than trends in extensive properties (amounts). Scatter plots would be more useful.*

We have converted Figure 1 of the original manuscript to a scatter plot and moved the original Figure 1 to the Supplementary Information.

**(21)** *L264: "similarly" - actually Babs and Bscat appear poorly correlated.*

We have rephrased this sentence.

**(22)** *L273 - please include the uncertainty on this value.*

A discussion has been in the introduction about the MAC values of pure, uncoated BC stressing that its value at a specific wavelength depends both on the size distribution of the particles but also on their degree of aging (fresh aggregates versus collapsed more spherical structures). The corresponding uncertainty is now discussed at this point of the paper.

**(23)** *L286: Not clear how coating thicknesses were constrained.*

Two different approaches are now used for the calculation of the coating thicknesses. In the first approach used in the original paper the coating of the BC core was calculated using the PM1 mass and the rBC distribution as it was measured by the SP2. This approach corresponds to the internal mixing assumption. In order to provide constraints in our method we also calculated in the revised paper the coating thickness of the rBC cores using the leading-edge-only (LEO) fit method (Gao et al. 2007) of the SP2 data. The thicknesses as expected were lower than those resulting from the internal mixing assumption, but were still significant. These methods are now explained clearly in the revised paper.

**(24)** *L281: AAE only described for 523 to 405. . . how the range calculated, averaged?*

The AAE was calculated by a power-law fitting of the babs measured by the aethalometer in all wavelengths (370, 470, 520, 590,660, 880, 950 nm). The information on the calculation of the AAE has been added in the revised manuscript.

**(25**) *L289: is 13 percent meaningful in the context of uncertainties?*

The new uncertainty analysis focuses on the robustness of this conclusion. Please note that the average was modest but there were periods of significant unexplained absorption.

**(26)** *L313 - I think you mean "no periods of enhanced BB influence"*

We have rephrased this sentence.

**References**

Arnott, W.P., Moosmüller, H., Sheridan, P. J., Ogren, J. A., Raspet, R., Slaton, W.V., Hand, J. L., Kreidenweis, S. M. and Collett Jr., J. L.: Photoacoustic and filter-based ambient aerosol light absorption measurements: instrument comparison and the role of relative humidity, J. Geophys. Res., 108, 4034–44, 2003.

Bond, T. C., and Bergstrom, R. W.: Light absorption by carbonaceous particles: An investigative review, Aerosol Sci. and Technol., 40, 27–67, 2006.

Bond, T. C., Habib, G., and Bergstrom, R. W.: Limitations in the enhancement of visible light absorption due to mixing state, J. Geophys. Res., 111, D20211, 2006.

Cappa, C., Lack, D., Burkholder, J., and Ravishankara, A.: Bias in filter based aerosol light absorption measurements due to organic aerosol loading: Evidence from laboratory measurements, Aerosol Sci. Technol., 42, 1022–1032, 2008.

Gkatzelis, G. I., Papanastasiou, D. K., Florou, K., Kaltsonoudis, C., Louvaris, E. and Pandis, S. N.: Measurement of nonvolatile particle number size distribution, Atmos. Meas. Tech., 9, 103–114, 2016.

Lack, D. A., Langridge, J. M., Bahreini, R., Cappa, C. D., Middlebrook, A. N., and Schwarz, J. P.: Brown carbon and internal mixing in biomass burning particles, Proc. Natl. Acad. Sci. USA, 109, 14802–14807, 2012.

Lack, D. A., and Cappa, C. D.: Impact of brown and clear carbon on light absorption enhancement, single scatter albedo and absorption wavelength dependence of black carbon, Atmos. Chem. Phys., 10, 4207–4220, 2010.

Lack, D. A., Cappa, C. D., Covert, D. S., Baynard, T., Massoli, P., Sierau, B., Bates, T. S., Quinn, P. K., Lovejoy, E. R., and Ravishankara, A. R.:Bias in filter based aerosol light absorption measurements due to organic aerosol loading: Evidence from ambient measurements, Aerosol Sci. Technol., 42, 1033–1041, 2008.

Lee, B. H., Kostenidou, E., Hildebrandt, L., Riipinen, I., Engelhart, G. J., Mohr, C., Decarlo, P. F., Mihalopoulos, N., Prévôt, A. S. H., Baltensperger, U., and Pandis, S. N.: Measurement of the ambient organic aerosol volatility distribution: Application during the Finokalia Aerosol Measurement Experiment (FAME-2008), Atmos. Chem. Phys., 10, 12149–12160, 2010.

Liu, C., Chung, C. E., Yin, Y., Schnaiter, M.: The absorption Ångström exponent of black carbon: from numerical aspects, Atmos. Chem. Phys., 18, 16409–16418, 2018.

Louvaris, E. E., Florou, K., Karnezi, E., Papanastasiou, D. K., Gkatzelis, G. I., and Pandis, S. N.: Volatility of source apportioned wintertime organic aerosol in the city of Athens, Atmos. Environ.,158, 138-147, 2017.

[Figure]

Ulbrich, I. M., Canagaratna, M. R., Zhang, Q., Worsnop, D. R., and Jimenez, J. L.: Interpretation of organic components from positive matrix factorization of aerosol mass spectrometric data, Atmos. Chem. Phys. 9, 2891–2918, 2009.
* * *

---

## Author Response (AR1)

**Responses to the Comments of Reviewer 1**

(1) The authors measure BC concentrations using an SP2. Not shown are any size distributions, which are important to consider as the measurements may have a negative bias owing to BC particles that are outside the detection window. This contribution may be small, but it should be considered if it has not been (it is not clear from the presentation). One way this has been dealt with in the literature is through single- or multi-modal fitting. This is another reason that the observed MAC might be higher than an expected value for pure BC (see line 278). This links to a question about measurement uncertainties, which are not reported. The authors must provide a discussion of uncertainties, that includes potential biases owing to factors such as BC outside of the measurement size window.

The BC number concentration distributions were fitted in our baseline calculations using a Gaussian distribution to account for particles smaller than the SP2 detection limit (Ditas et al., 2018). This extrapolation resulted in an increase of the BC mass concentration by 3-8%, so its effect on the reported results (e.g., the observed MAC) was minor. This information has been added to the revised paper together with a graph (in the Supplementary Information) with a few representative mass distributions. A discussion of other SP2 uncertainties has been also added (Lack et al., 2012). This SP2 issue is discussed together with the rest of the uncertainties of both the measurements and the modeling in a new uncertainty analysis section that has been added to the revised paper.

(2) A major concern that I have about this paper relates to their interpretation of their absorption measurements. They state that the MAC of pure, uncoated BC at 405 nm should be  $9.9 \text{ m}^2/\text{g}$ , but that they observe an average of  $16.3 \text{ m}^2/\text{g}$ . However, they conclude later that their predicted value of  $14.1 \text{ m}^2/\text{g}$  is a factor of two greater than it should be. This implies a MAC of  $7.05 \text{ m}^2/\text{g}$  for pure BC from the calculations, much smaller than the  $9.9 \text{ m}^2/\text{g}$ . This is because Mie theory generally leads to underestimates of the BC absorption. Thus, the calculated MAC is biased low. The extent of the underestimate depends on the details of the calculations (refractive index, particle size), which are not provided. The amount of information regarding the calculations is insufficient to allow clear judgement of their appropriateness. Regardless, it is evident that the authors are not making a fair comparison; the  $14.1 \text{ m}^2/\text{g}$  value cannot be compared to the  $16.3 \text{ m}^2/\text{g}$  value as they are starting from different reference values. Thus, the authors conclusion on L290 that there must be non-refractory absorbing material is not justified, nor are any subsequent calculations of the "delta\_MAC". Related, the conclusions and calculations regarding the brown carbon imaginary RI, as determined.

This is a valid point but it is due to a misunderstanding of this rather poorly placed sentence. Our calculations are self-consistent. The MAC of 9.9  $m^2/g$  was never used in any of the calculations and should not be part of the discussion here. It was mentioned at this point just to support the argument that the measured value of 16.3  $m^2/g$  is too high compared to what would be expected for pure uncoated BC. We have moved the discussion of the MAC values of pure, uncoated BC to the introduction of the paper stressing that its value at a specific wavelength depends both on the size distribution of the particles but also on their degree of aging (fresh aggregates versus collapsed

more spherical structures). We just make the point now that the measured value of  $16.3 \text{ m}^2/\text{g}$  is far outside the range expected for pure, uncoated BC particles.

The details of all the Mie theory calculations are now provided at one place in the paper. Some of them were in different parts of the original work. The predicted average BC MAC405 using the measured BC size distribution and assuming pure uncoated spherical particles was 7.2  $m^2 g^{-1}$  varying from 6.1 to 7.8  $m^2 g^{-1}$  during the study (hourly averages). This is consistent with the expected 5.3-9.2  $m^2 g^{-1}$  for BC core diameters in the 10-350 nm size range. This information has been also added to the paper.

We should also stress that the 16.3  $m^2/g$  is the average measured value based on the PAX and the SP2 measurements. It is not based on an assumed reference value for the pure, uncoated BC MAC value. Thus, there is no issue regarding the consistency of the predicted and the measured values. The uncertainty of both is analyzed in the new uncertainty analysis section. Please note that the differences of the measured and the predicted MAC405 were as high as 7 m2/g and we argue that this difference cannot be explained by the corresponding uncertainties in the measurements and the modeling.

(3) I do not find that the inputs to their calculations are sufficiently constrained to allow for accurate calculation of the absorption. Certainly, their calculated value cannot be directly compared to the observations (as the authors do) owing to my point in comment 2 that they undoubtedly underestimate the absorption by BC in their calculations. Beyond that, the authors seem to make a very poor assumption regarding the coatings on BC. The note that they use the ratio of the total aerosol mass divided by BC mass to estimate coating thickness. This is not appropriate. The BC-containing particles likely make up a small fraction of the total particles. Thus, some unknown fraction of the total non-BC material is internally mixed with BC. Quite often this fraction can be quite small, although it depends explicitly on the history of the air mass. If the authors are unable to provide constraints on the actual amount of coating on their BC, their calculated Eabs must be taken as an upper limit, with the actual value falling somewhere between 1 and the upper limit (2.07); unfortunately, no tighter constraint is possible without additional information regarding the true coating state. It is evident that the assumption that all material coats BC is a poor assumption, as the variability in the calculated MAC is negligible; this is simply because the particles always have "thick" coatings, in the plateau of the Mie curve, and thus little variability in the calculated MAC. The calculated MAC cannot be compared to the observations in any sort of quantitative manner. The conclusion that the lensing-induced enhancement is 2.07 is not justified. The authors might consider comparing the observed MAC 405 versus the NRPM to BC ratio; if the authors assumption that there is a substantial absorption enhancement and that all NRPM material is coated on BC were valid they should see a strong relationship between the MAC 405 and the NRPM/BC ratio that trends towards the expected pure BC value as the NRPM/BC ratio declines. (There may still be a relationship between the coating amount and the total NRPM/BC ratio, and thus even if a relationship is observed it is not definitive proof. Nonetheless, it might provide guidance for the interpretation.)

The uncertainty of the estimated value of the absorption of the BC is addressed in the new uncertainty analysis section where we try additional values of the BC refractive index (both high and low). We test indirectly the hypothesis that this absorption has been underestimated and that

this can explain the difference in measured and predicted absorption of the particles. We show that while increasing the absorption of the BC cores can explain the average observed absorption it cannot explain its variability. The link between this variability and the ELVOC concentrations remains robust for both the low and high values of the refractive index.

The second point of the reviewer is that our calculations are based on the assumption that all particles of the same size contain the same BC (internally mixed aerosol population). This is correct and it does provide the upper limit for the lensing effect and therefore the minimum of the unexplained absorption that could be assigned to BrC. All the air masses sampled during the campaign were quite aged so it is reasonable to assume that all the BC particles were coated. In order to provide a better constraint for our method we calculated the coating thickness of the rBC cores using the leading-edge-only (LEO) fit method (Gao et al. 2007) and the SP2 data. The thicknesses as expected were lower than those resulting from the internal mixing assumption but were still significant. We have repeated our calculations using this estimate of the coating thickness and tested the robustness of the link between the unexplained absorption and the ELVOC concentrations but also for the rest of our results.

We have followed the reviewer's suggestion and investigated the relationship between MAC405 and the ratio of non-refractory PM1 and rBC. There is a positive correlation between the two (R=0.31) and the intercept is 12.8 m2/g. This analysis provides, as the reviewer suggests, limited support to our argument, but has been added to the paper nonetheless.

(4) The authors report a strong correlation between the MAC at 405 nm and the BC concentration. They do not report the relationship at other wavelengths. If BrC is making a major contribution, and if it does not come from primary emissions that are associated with the BC emissions, then there should be a stronger relationship at longer wavelengths. The authors have measurements at longer wavelengths from the Aethalometer. Is the  $R^2$  value larger at longer wavelengths than it is at shorter wavelengths?

We followed the reviewer's suggestion and examined the values of the  $R^2$  between the MAC and the rBC concentration at different wavelengths. Indeed, the  $R^2$  at 370 nm was 0.67, a value lower than the  $R^2$  at 405 nm (equal to 0.74). This is consistent with the presence of some BrC. However, the first is based on the Aethalometer and the second on the PAX measurements. Moving at higher wavelengths the  $R^2$  at 450 nm was 0.68, at 520 nm 0.67, at 590 nm 0.65, and at 660 nm 0.65. These values provide a mixed picture. Part of the explanation for this behavior could be that a fraction of the BrC is associated with rBC either from the emissions or from the associated gas-phase pollutants that react in their way to the site. This information has been added to the revised paper and the corresponding figures to the Supplementary Information.

(5) One of the conclusions of Karnezi et al. (2014) is that "Our results indicate that existing TDbased approaches quite, often cannot estimate reliably the OA volatility distribution, leading to large uncertainties, since there are many different combinations of the three properties that can lead to similar thermograms." Yet, this is exactly what the authors have done here. It is thus unclear how the authors know that they have a unique solution, especially with respect to the co-variation between the derived volatility distribution and accommodation coefficient. This is a valid point. The comment of Karnezi et al. (2014) referred to a detailed volatility distribution in the Volatility-Basis-Set framework. We have recently shown (Louvaris et al., 2017; Cain et al., 2020) that the uncertainty of volatility classes (e.g. ELVOCs, LVOCs, SVOCs) is significantly less than the uncertainty of individual volatility bins. The uncertainty of these concentrations is provided by the fitting algorithm used in the paper and are shown in Figure 4 of the paper. We have repeated the analysis of the links between the unexplained absorption and the ELVOC levels. The results are discussed in the new uncertainty analysis section and show that our original conclusion is robust.

(6) Why would one expect that an intensive property (the delta\_MAC) should correlate with an extensive property (the ELVOC concentration)? Typically, intensive properties should correlate with some other intensive property (for example, the ELVOC fraction of total OA). Extensive properties should correlate with extensive properties (for example, the estimated unexplained absorption vs the absolute ELVOC concentration). I do not see a justification for why an intensive measurement should correlated with an extensive. The authors highlight this relationship in their abstract, yet spend a total of 7 lines presenting and discussing it. This is, in my opinion, insufficient. Does the delta\_MAC correlate with any other measured properties, as but one thing that it would be useful for the authors to consider? That said, I'll note again that I do not believe the delta\_MAC values are valid for the reasons discussed above, and thus, in my opinion, this entire analysis is suspect.

This is a valid point and we agree with the reviewer that it is preferable to correlate either intensive properties or extensive properties to each other. We have examined the correlation of the unexplained  $b_{abs}$  and the ELVOC concentration and found an R2=0.76. This actually strengthens our argument regarding the contribution of the ELVOCs to the unexplained absorption by removing the other factors entering into the calculation of the intensive property the MAC. We have revised Figure 7 accordingly.

Please note that we devote two pages in the original manuscript explaining the estimation of the ELVOC concentrations and a few more pages discussing the calculation of the unexplained absorption.

We have examined the correlations of  $\Delta$ MAC with the rBC, with other parameters and found that they were all much lower than those for the ELVOCs (0.76). For example the  $R^2$  with the rBC was 0.38 (rBC), with the LO-OOA 0.44 and with the MO-OOA 0.29. There was a relatively high  $R^2$  with the sulfate levels (0.59) that could be interesting as high sulfate levels in this area correspond to high aerosol acidity which has been shown to promote formation of oligomers in secondary organic aerosol. This analysis has been added to the paper.

(7) Also, what might be the source of these low-volatility absorbing organic components in this environment? This is not discussed. Why would the low-volatility components from LO-OOA have the same absorptivity as those from MO-OOA, as implied in the equation at line 300. It is generally thought that OA from biogenic sources is relatively non-absorbing. Certainly, I am unaware of any measurements that indicate that SOA from biogenics, which presumably dominate this site, can have an RI as large as 0.4, which the authors indicate was measured. The authors need to provide justification via comparison to the literature.

The OA in the site is the result of long range transport from other areas and is not due to local sources (including biogenics). During the period of the study the area near the station has almost no vegetation (mainly dried out bushes). The average O:C of the OA in the site (around 0.83 on average) during the campaign is a lot higher than that of fresh biogenic SOA. Air mass came to the site from continental Greece and Balkans, Turkey, Africa and central Mediterranean. The OA in the Eastern Mediterranean during the summer has significant contributions from both anthropogenic and biogenic sources but their contributions remain uncertain. Based on our recent work (Drosatou et al., 2019) the LO-OOA and MO-OOA do not reflect different sources, but rather different degrees of chemical aging. ELVOCs can be produced both by primary sources (combustion of fossil fuels but also biomass burning) and secondary processes. Given the intense chemical processing of the organic compounds from all sources in their way to Finokalia, we cannot assign based on our measurements to a specific source or sources. The association of BrC with material of lower volatility away from its sources has been reported in a number of studies focusing on biomass burning (Saleh et al., 2014; Wong et al., 2019). This discussion has been added to the revised paper.

(8) The authors measured light absorption coefficients using an aethalometer and a PAX. As best I can tell, the authors have not compared the measurements from these two instruments to establish whether the aethalometer suffers from any positive biases that are known to impact filter-based absorption measurements. That said, such biases are less likely to influence the wavelength dependence measurements than they are the absolute absorption values.

A comparison of the aethalometer and the PAX measurements has been added to the revised paper including two new figures in the Supplementary Information. We compared the absorption coefficients measured by the aethalometer at 370 nm and at 450 nm with that measured by the PAX at  $\lambda$ =405 nm. The measurements of the two instruments were highly correlated with R2=0.9-0.91. The babs,370 measured by the aethalometer was higher than the babs,405 of the PAX. Their relationship is described by the equation y=3.32x - 3.24. Similarly, the babs,450 measured by the Aethalometer was higher than that of the PAX at 405 nm and their relationship was described by the equation y=1.85x - 0.93. These discrepancies could possibly exist due to the artifacts associated with OA loadings on the filter (Cappa et al., 2008; Lack et al., 2008). A brief discussion has been added in the main paper and the corresponding figures have been added to the SI.

We have added a new section in the paper describing our uncertainty analysis of the results. This takes into account both the uncertainty of the measurements but also the uncertainty introduced by the various assumptions during the theoretical analysis. We have followed the reviewer's suggestion and addressed the uncertainty of the PMF analysis. We used the bootstrapping approach (Ulbrich et al., 2009) performing ten additional simulations. The estimated uncertainty for the LO-OOA concentrations was 2% and for the MO-OOA concentrations was 3%. These relatively small uncertainties suggest that the PMF uncertainty regarding the determination of these factors does not affect our conclusions. The corresponding results have been added to the Supplementary Information and are discussed in the new uncertainty analysis section. We have also looked at the correlations between the concentrations of MO-OOA, LO-OOA and BC. The  $R^2$  between the hourly concentrations of MO-OOA and rBC was 0.12, and for LO-OOA and rBC 0.17. This is not unexpected given that Finokalia is far away from the corresponding sources of both BC and organic compounds and significant physical and chemical processing has taken place during the

transport of the aerosol from the sources to the receptor. The  $R^2$  between the hourly concentrations of the ELVOCs and rBC was 0.25. This information has been added to the revised paper.

**(3) Some issues have not been sufficiently addressed. They include: a) how is the observation of AAE of $\sim$ 1 consistent with expectations for significant BrC absorption? In fact, can any significant absorption be attributed to low volatility species?**

The observation of an average AAE ~ 1 is not inconsistent with the presence of some BrC especially during selected periods. For example, Lack and Cappa (2010) argued that an AAE<1.6 does not exclude the possibility of BrC; rather BrC cannot be confidently assigned unless AAE>1.6. The AAE of coated BC is highly sensitive to particle size distribution and it decreases as particle size increases (Liu et al., 2018). This last study shows that relatively low values of AAE should be expected in a site like Finokalia and demonstrated the importance of various parameters on the BC AAE and the potential problems introduced by assuming BC AAE as being equal to 1.0. Finally, please note that based on our estimates BrC led to a 15% increase of the campaign average  $b_{abs}$  at 405 nm. This increase is on average modest and is not inconsistent with the relatively low AAE observed in this remote site with relatively large, heavily processed particles. The finding that this unexplained absorption increases as the concentration of the ELVOCs increases is probably the most interesting finding of our work. We have added discussion of these points in the revised paper.

**(4) b) how is absorption of internal mixtures with rBC separated from bulk phase absorption?**

The assumption in our baseline analysis is that the non-refractory absorbing material is coating the BC particles. All the air masses sampled during the campaign were quite aged so it is reasonable to assume that all the BC particles were coated. Our baseline calculations are based on the assumption that all particles of the same size contain the same BC (internally mixed aerosol population). This provides the upper limit for the lensing effect and therefore the minimum of the unexplained absorption that could be assigned to BrC. We have performed an additional calculation using the coating estimated by the SP2 measurements to test the robustness of our conclusions. Additional assumptions are examined in the added uncertainty analysis section.

**(5) c) How is the PMF validated? Did rBC concentration (not deviation in MAC) correlate or anticorrelate with either factor?**

We used the bootstrapping approach (Ulbrich et al., 2009) performing ten additional simulations. The estimated uncertainty for the LO-OOA concentrations was 2% and for the MO-OOA concentrations was 3%. These relatively small uncertainties suggest that the PMF uncertainty regarding the determination of these factors does not affect our conclusions. Both MO-OOA and LO-OOA were positively correlated with rBC with the corresponding correlation coefficient R being 0.35 for MO-OOA and 0.41 for LO-OOA. These correlations between the absolute concentrations of the various PM components are in general expected in remote sites that are affected sometimes by more polluted air masses (leading to increases of the major PM components) and cleaner air masses processed by rain (leading to decreases of the major PM components. For example, the R of the non-refractory PM1 and the rBC was 0.55. The

corresponding results have been added to the Supplementary Information and are discussed in the new uncertainty analysis section.

**(6) d) How are coating thicknesses constrained for rBC?**

Two different approaches are now used for the calculation of the coating thicknesses. In the first approach used in the original paper the coating of the BC core was calculated using the  $PM_1$  mass and the rBC distribution as it was measured by the SP2. This approach corresponds to the internal mixing assumption. In order to provide constraints in our method we also calculated in the revised paper the coating thickness of the rBC cores using the leading-edge-only (LEO) fit method (Gao et al. 2007) of the SP2 data. The thicknesses as expected were lower than those resulting from the internal mixing assumption, but were still significant. We have repeated our calculations using this estimate of the coating thickness and tested the robustness of the link between the unexplained absorption and the ELVOC concentrations but also for the rest of our results.

(7) e) How high is confidence, based on measurement and theoretical uncertainties, that the range of MAC is not solely associated with 1) rBC and PAS uncertainties or 2) lensing enhancements? An uncertainty analysis has been added to the revised paper to address these issues. The PAX has an uncertainty of less than 10% for the aerosol absorption measurement (Lack et al. 2012). Part of the uncertainty of the rBC mass by the SP2 is related to the detection limit of the smaller BC particles. The BC distributions were fitted in our baseline calculations using a Gaussian distribution to account for particles smaller than the SP2 detection limit (Ditas et al., 2018). This extrapolation resulted in an increase of the BC mass concentration by 3-8%, so its effect on the reported results (e.g., the observed MAC) was minor. The average propagated uncertainty of the MAC405, based on the measurements of rBC and babs at  $\lambda$ =405 nm is approximately 20%. Mie theory calculation can underestimate the absorption due to BC morphology and mixing. These uncertainties are discussed together with the rest of the uncertainties of the revised paper.

**(8) The reader needs more information about what was done in order to understand the results: a) How did the PAS and Aethalometer compare? Which was used to generate MAC405? How significant were the differences between them?**

A comparison of the aethalometer and the PAX measurements has been added to the Supplementary Information of the revised paper. We compared the absorption coefficients measured by the aethalometer at 370 nm and at 450 nm with that measured by the PAX at  $\lambda$ =405 nm. The measurements of the two instruments were highly correlated with R2=0.9-0.91. The babs,370 measured by the aethalometer was higher than the babs,405 of the PAX. Their relationship is described by the equation y=3.32 x - 3.24. Similarly, the babs,450 measured by the Aethalometer was higher than that of the PAX at 405 nm and their relationship was described by the equation y=1.85x - 0.93. These discrepancies could possibly exist due to the artifacts associated with OA loadings on the filter (Cappa et al., 2008; Lack et al., 2008). A brief discussion has been added in the main paper and the corresponding figures have been added to the SI. The MAC405 was calculated by dividing the babs,405 measured by the PAX and the rBC mass. This point is clarified in the revised manuscript to avoid confusion.

(9) b) SP2 - laser intensity? size range of detection for rBC? Correction to account for total rBC accumulation mode concentrations? Type (8 channel?). Precision of cal with CPMA? Total uncertainty estimate in concentration (noting that different BC does have different response in SP2)? Calculation of coating thicknesses?

The SP2 uses a Nd:YAG Laser at 1064 nm operated at 4 V and 3400 A. The calibration of the SP2 was verified in separate experiments using a centrifugal particle mass analyzer (CPMA, Cambustion). The SP2 mode mass and the CPMA mode mass were in good agreement,  $R^2$ =0.99 (Saliba et al., 2016). This information has been added to the experimental section.

The BC number concentration distributions were fitted in our baseline calculations using a Gaussian distribution to account for particles smaller than the SP2 detection limit (Ditas et al., 2018). This extrapolation resulted in an increase of the BC mass concentration by 3-8%, so its effect on the reported results (e.g., the observed MAC) was minor. This information has been added to the revised paper together with a graph (in the Supplementary Information) with a few representative mass distributions. A discussion of other SP2 uncertainties has been also added (Lack et al. 2012). This SP2 issue is discussed together with the rest of the uncertainties of both the measurements and the modeling in a new uncertainty analysis section that has been added to the revised paper.

Two different approaches are now used for the calculation of the coating thicknesses. In the first approach used in the original paper the coating of the BC core was calculated using the  $PM_1$  mass and the rBC distribution as it was measured by the SP2. This approach corresponds to the internal mixing assumption. In order to provide constraints in our method we also calculated in the revised paper the coating thickness of the rBC cores using the leading-edge-only (LEO) fit method (Gao et al. 2007) of the SP2 data. The thicknesses as expected were lower than those resulting from the internal mixing assumption, but were still significant. We have repeated our calculations using this estimate of the coating thickness and tested the robustness of the link between the unexplained absorption and the ELVOC concentrations but also for the rest of our results.

(10) c) Sampling: was the aerosol dried? What temperature was the laboratory? If it was not dried, the whole analysis is likely questionable. How large were the corrections for diffusion and thermophoresis losses in the denuder (I expect likely negligible for Accum. mass).

A diffusion drier was used upstream of the optical measurements to reduce any relative humidity-related measurement artifacts in the measurements (Arnott et al. 2003). The campaign average temperature and relative humidity were  $22\pm4$  °C and  $53\pm19$  %, respectively. The measurement station had an AC unit maintaining the temperature at 25 °C.

To account for losses in the thermodenuder, sample flow rate as well as size- and temperature-dependent loss corrections were applied following Louvaris et al. (2017) corresponding to the operating conditions during the campaign. Less than 20% of the particulate matter was lost in the TD, at temperatures up to 100°C, while the losses increased for higher temperatures. For a temperature equal to 400 °C almost 50% of particles larger than 50 nm

is lost. The uncertainty introduced by the correction ranged was approximately 20% (Gkatzelis et al., 2016; Louvaris et al., 2017).

The above points have been added to the manuscript.

(11) d) Mie theory tests: what constraints on coating thickness were used in the Mie Theory evaluation of index of refraction? Was this derived from SP2 data? How? This is a critical question if there is to be any conclusion drawn from this analysis. Note that SP2 determination of coating thicknesses are highly uncertain for thin coatings (e.g. Ohata et al. (2016). Hygroscopicity of materials internally mixed with black carbon measured in Tokyo, J. Geophys. Res. Atmos., 121, doi:10.1002/2015JD024153.)

Two different approaches are now used for the calculation of the coating thicknesses. In the first approach used in the original paper the coating of the BC core was calculated using the  $PM_1$  mass and the rBC distribution as it was measured by the SP2. This approach corresponds to the internal mixing assumption. In order to provide constraints in our method we also calculated in the revised paper the coating thickness of the rBC cores using the leading-edge-only (LEO) fit method (Gao et al. 2007) of the SP2 data. The thicknesses as expected were lower than those resulting from the internal mixing assumption, but were still significant. We have repeated our calculations using this estimate of the coating thickness and tested the robustness of the link between the unexplained absorption and the ELVOC concentrations but also for the rest of our results. A coating refractive index was assumed of 1.55 following previous studies (Bond and Bengstrom, 2006).

(12) Line-level and specific comments L37 - averaged over a population. The literature review 34-83 - lensing enhancement is well written, but might give the reader the incorrect idea that similar work was done here. As only the AMS and SMPS were behind denuder this is not the case. (Note that it is a nice feature of your data set that initial brown carbon bleaching/evap should have already substantially occurred).

In the literature review we wanted to summarize the finding of previous studies but also the different techniques used. We clarify that in our study only the AMS and the SMPS were behind the thermodenuder and that an interesting feature of the dataset is that the organic aerosol has been thoroughly processed.

(13) TD - please provide more info about the temperature set points so that the reader isn't surprised by the lack of data between 200 and  $\sim$ 375 °C.

We have added the detailed information about the temperature steps used in the thermodenuder explaining the change from 200 to approximately 375 C.

(14) L197 Bond and Bergstrom, 2006 suggest (1.95,-0.79) for the complex index at 1.8 g/cc. Not as stated.

For the BC core we assumed a refractive index of the core  $n_{rBC}=1.85+0.71i$  based on Bond et al. (2006). We have corrected the reference in the revised manuscript.

(15) The paper by China et al., 10.1002/2014GL062404 will be useful for justifying use of Mie theory shell and core.

We have added the suggested reference to the revised paper to further support the use of core and shell Mie theory for our site.

**(16) Please consider if Figure 2 is unnecessary given Figure 4a? Combine?**

We believe that the existence of Figure 2 is helping the reader for easy comparison between our study and previous studies in area (Lee et al., 2010). In addition, the plot is needed for comparison with other studies that use similar methods to calculate the OA volatility. Finally, the use of both Figure 2 and Figure 4 show how much each factor MO-OOA, and LO-OOA affects the total OA volatility.

(17) Wavelength of Mie theory calcs? Note that RI for BC is likely low. Were the calculations carried out for observed core size dist? coating thickness? On what time scale were those calculated?

The Mie theory calculation was performed at a wavelength of 405 nm in order to be able to compare our results with the measurements from the PAX. The sensitivity to the refractive index of BC was investigated and the results are discussed in the added uncertainty analysis section. The calculations were carried out based on the measured rBC core size distribution extrapolated so smaller sizes. Two different coating thicknesses were used. One based on the internal mixture assumption providing an upper limit for the calculation and one based on the thicknesses calculated by the SP2. One-hour averaged data were used as inputs in the model. The above information has been added to the revised paper.

(18) Figure 7: negative Measure-Predicted not included - meaningless without all the data. It appears that some positive M-P data is also missing (based on the number of data points in figure 7 vs those in Figure 6a).

Please note that the data in Figure 6a are one-hour averages while those in Figure 7 are 3-hour averages. This is now explained in the corresponding figure captions. The original Figure 7 has been redrawn using the unexplained absorption in the y-axis.

(19) L249: "average calculated saturation concentration of  $0.016 \text{ ug/m}^3$ ". Corrected.

(20) Time series not very helpful for understanding more than trends in extensive properties (amounts). Scatter plots would be more useful.

We have converted Figure 1 of the original manuscript to a scatter plot showing the individual measurement points.

(21) L264: "similarly" - actually Babs and Bscat appear poorly correlated. We have rephrased this sentence.

(22) L273 - please include the uncertainty on this value.

A discussion has been in the introduction about the MAC values of pure, uncoated BC stressing that its value at a specific wavelength depends both on the size distribution of the particles but also

on their degree of aging (fresh aggregates versus collapsed more spherical structures). The corresponding uncertainty is now discussed at this point of the paper.

**(23) L286: Not clear how coating thicknesses were constrained.**

Two different approaches are now used for the calculation of the coating thicknesses. In the first approach used in the original paper the coating of the BC core was calculated using the  $PM_1$  mass and the rBC distribution as it was measured by the SP2. This approach corresponds to the internal mixing assumption. In order to provide constraints in our method we also calculated in the revised paper the coating thickness of the rBC cores using the leading-edge-only (LEO) fit method (Gao et al. 2007) of the SP2 data. The thicknesses as expected were lower than those resulting from the internal mixing assumption, but were still significant. These methods are now explained clearly in the revised paper.

**(24) L281: AAE only described for 523 to 405... how the range calculated, averaged?**

The AAE was calculated by a power-law fitting of the  $b_{abs}$  measured by the aethalometer in all wavelengths (370, 470, 520, 590, 660, 880, 950 nm). The information on the calculation of the AAE has been added in the revised manuscript.

**(25) L289: is 13% meaningful in the context of uncertainties?**

The new uncertainty analysis focuses on the robustness of this conclusion. Please note that the average was modest but there were periods of significant unexplained absorption.

**(26) L313 - I think you mean "no periods of enhanced BB influence"**

We have rephrased this sentence.

**References**

[revised manuscript text omitted]

Commented [AT2]: REVIEWER 2 Comment 9

Commented [AT3]: REVIEWER 2 Comment 9

Commented [AT4]: REVIEWER 1 comment 1 ; Reviewer 2 comment 9

Commented [AT5]: REVIEWER 2 Comment 7

diffusion drier was used upstream of the optical measurements. The campaign average temperature and relative humidity were  $22\pm4$  °C and  $53\pm19$  %, respectively. The measurement station had an AC-unita temperature-control system maintaining the temperature at approximately 25 °C. A diffusion drier was used upstream of the optical measurements.

The thermodenuder (TD) used in this study, was placed upstream of the HR-ToF-AMS and the SMPS. The TD design was similar to that developed by An et al. (2007) and is described by Louvaris et al. (2017). The TD was operated at temperatures ranging from 25 °C to 400 °C using several temperature steps from 25 °C to 200 °C over several hours and then rapidly (in 20 min) increasing its temperature to the 375-400,°C to investigate the presence of ELVOCs. One complete cycle from 25 to 400 °C and back to 25 °C lasted approximately 10 h. The temperature was increasing from ambient to 2009C for approximately 6 hours. Then the temperature was increased to a higher temperature 4006C in order to investigate the existence of ELVOCs. The increase from 200.0C to 4000C was approximately 20 min. Sampling was alternated between the ambient line and the TD line every 3 minutes using computer-controlled valves. Changes in particle mass concentration, composition, and size due to evaporation in the TD were measured by the HR-ToF-AMS and the SMPS resulting in thermograms of OA mass fraction remaining (MFR) as a function of TD temperature. The OA MFR was calculated as the ratio of organic mass concentration of a sample passing through the TD at time  $t_i$  over the average mass concentration of the ambient samples that passed through the bypass line at times  $t_{i-1}$  and  $t_{i+1}$ . The sample residence time in the centerline of the TD was 14 s at 25 °C, corresponding to an average residence time in the TD of 28 s. The MFR values were corrected for particle losses in the TD due to diffusion and thermophoresis. To account for these losses, sample flow rate as well as size- and temperaturedependent loss corrections were applied following Louvaris et al. (2017) corresponding to the operating conditions during the campaign. Less than Up to 20% of the particulate matter was lost in the TD7 at temperatures up to 100°C.7 Twhile the losses increased atfor higher temperatures. For a temperature equal to and at 400 °C approximately almost 50% of particles ulates larger than 50 nm wasis lost. The uncertainty introduced by the loss correction was approximately ranged from 20-to 30% (Gkatzelis et al., 2016; Louvaris et al., 2017). The final step of the data analysis was to average the corrected for CE and TD losses MFR data based on temperature bins of 10°C. The MFR calculation assumes implicitly that the OA concentration remains constant during the measurement period. To ensure that this condition is satisfied, if two consecutive OA ambient

Commented [AT6]: REVIEWER 2 comment 10

| Commented [SP7]:                          |  |
|-------------------------------------------|--|
| Commented [SP8R7]: REVIEWER 2, Comment 12 |  |
|                                           |  |
| Formatted: Superscript                    |  |
| Formatted: Superscript                    |  |
| Formatted: Superscript                    |  |

Commented [AT9]: REVIEWER 2 comment 13

Commented [AT10]: REVIEWER 2 comment 10

[revised manuscript text omitted]

Commented [SP11]: REVIEWER 2 Comment 17

Commented [AT12]: REVIEWER 2 comment 15

Commented [AT13]: REVIEWER 2 comment 14

Commented [SP14]: REVIEWER 2 Comment 23

Commented [SP15]: REVIEWER 1, Comment 2

distribution. The other inputs of the model were the measured BC size distributions from the SP2 and the ratio of the total aerosol mass over the BC mass as measured. by the SMPS and the SP2, respectively.\_1-hour averaged data were used as inputs in the model. The size of the core is assumed based on the rBC size distribution. The total aerosol mass divided by BC was used to estimate the coating thickness based on the assumption that the BC material is internally mixed with the non-refractory aerosol species (Saliba et al., 2016).

The assumption of internal mixture can be justified by the lack of any local sources. All the air masses sampled during the campaign were quite aged, so it is reasonable to assume that all the BC particles were coated. PMF analysis did not show any fresh emissions like HOA or BBOA and no acetonitrile was detected. FLEXPART analysis confirmed that the air masses measured were transferred from long distance areas (30% continental Greece and the Balkans, 13% Aegean, 24% Africa, 32% Italy/Sicily). The measurements also suggest internal mixture the relationship between MAC405 and the ratio of non refractory PM4 and rBC. There is a positive correlation between the two, (R=0.08) and the intercept is 12.1 m2 g-4 (Figure S10). In order to provide a better constraint for the analysismethod an additional calculation of the coating thickness of the rBC cores was also estimated using the leading-edge-only (LEO) fit method (Gao et al. 2007) and the SP2 data. The thicknesses as expected were lower than those resulting from the internal mixing assumption, but were still significant (Figure S11). Repeated Mie theory calculations using the coating thickness based on the LEO fit method are presented in the uncertainty analysis section.

The MAC values of pure, uncoated BC at a specific wavelength depend both on the size distribution of the particles but also on their degree of aging (fresh aggregates versus collapsed more spherical structures). A previous study has shown that pure BC core diameters in the 10-350 nm size range have an expected MAC550=3.9-6.8 m2-g-1-or MAC405=5.3-9.2 m2-g-1-(Bond et al., 2006). In this study the MAC405 predicted by Mie theory and the MAC405-calculated by the ratio of babs.405-over the rBC measured by PAX405 and SP2 were compared. The comparison was used to identify the absorption enhancement due to the lensing effect as well as the existence of an absorbing coating. Commented [SP16]: REVIEWER 1, Comment 2

Commented [AT17]: REVIEWER 1 Comment 3

**4. Results and discussion**

[revised manuscript text omitted]

The absorption coefficients measured by the aethalometer at  $\lambda$ =370 nm and at  $\lambda$ =470 nm, were compared with the absorption coefficient measured by the PAX at  $\lambda$ =405 nm. The measurements of the two instruments were highly correlated with *R*2=0.9-0.91. The babs,370 measured by the aethalometer was higher than the babs,405 of the PAX. Their relationship is Commented [AT19]: REVIEWER 2 comment 19

Commented [AT20]: REVIEWER 2 comment 21

described by the equation y=3.32x - 3.24. Similarly, the  $b_{abs,450}$  measured by the aAethalometer was higher than that of the PAX at 405 nm and their relationship was described by *the equation* y=1.85x - 0.93 (Figure S14). Part of these differences are due These discrepancies could possibly exist due to the artifacts associated with filter-based absorption measurements OA loadings on the filter-(Cappa et al., 2008; Lack et al., 2008).

Acetonitrile is a known biomass burning marker and can help identify the potential influence of the site by biomass burning events or wildfires during the campaign. The acetonitrile concentration measured by the PTR-MS remained close to 0.4 ppb during the campaign, which is the local background level. This together with the low BC levels indicate that the site was not impacted by nearby biomass burning during the study.

The  $b_{abs,405}$  variation followed that of the rBC (R2=0.74 for the hourly averages). The R2 at  $\lambda$ =370 nm was lower at 0.67 than that at 405 nm., a value lower than the R2 at  $\lambda$ = 405 nm. This is consistent with the presence of some BrC. However, the  $b_{abs,370}$  is measured by the aethalometer and the  $b_{abs,405}$  byon the PAX-measurements. For the  $b_{abs}$  at higher wavelengths measured by the aethalometer the R2 at  $\lambda$ =450 nm was 0.68, at  $\lambda$ =520 nm 0.67, at  $\lambda$ =590 nm 0.65, and at  $\lambda$ =660 nm 0.65 (Figure S15). These values are rather inconclusive regarding cannot justify the existence or absence of BrC. Part of the explanation for this behavior could be that a fraction of the BrC is associated with rBC either from the emissions or from the associated gas-phase pollutants that react in their way to the site.

The ratio of the  $b_{abs,405}$ , measured by the PAX, over the rBC mass, was equal to dMAC405=16.3±4.2 m2 g-1The ratio of the  $b_{abs,405}$  over the rBC, MAC405, was equal to 16.3±4.2 m2 g-1. Using the definition of the AAE3

$$AAE = -\frac{\ln\left[\frac{MAC_{405}}{MAC_{532}}\right]}{\ln\left[\frac{405}{277}\right]}$$

we can calculate the MAC of freshly generated BC 405 nm... In this study the average rBC size distribution had diameter ranged from 10 to 350 nm with a number mode diameter of an 67 nm and a mass mode diameter equal to mean at 185106 nm. TIt is clear that the measured MAC 405 is clearly higher bigger than the expected MAC 405 of uncoated BC particles (Bond et al., 2006). Previous studies have shown that freshly generated BC has a MAC 532 of 7.5 m2 g-1 (Clarke et al., 2004; Bond and Bergstrom, 2006). Using the definition of the AAE,

Commented [AT21]: Reviewer 1 Comment 8 ; Reviewer 2 comment 8

Commented [AT22]: REVIEWER 1 Comment 4

we can calculate the MAC of freshly generated BC 405 nm. Assuming that the AAE is equal to unity for BC particles, we find that the MAC405 of BC was approximately 9.9 m2g4. The difference between the measured and the theoretical value of MAC405 can be due to the coating of BC by other PM components (lensing effect) and/or the existence of other absorbing material. These two potential explanationsis will be explored in the following paragraphsbelow.

The AAE of the aerosol was calculated by using calculated by a power-law fitting of the  $b_{abs}$  measured by the aethalometer in all seven wavelengths (370, 470, 520, 590, 660, 880, 950) nm). The campaign average AAE of PM for wavelengths ranging from 370 to 950 nm was 0.97±0.22. Lack and Cappa (2010) suggested that an AAE>1.6 should confirm the presence of non-BC absorbing material, however an AAE<1.6 does not exclude its presence. The AAE of coated BC cores can deviate from the typical AAE=1 with values greater or lower than unity (Gyawali et al., 2009). In addition, Lack and Cappa (2010) suggested that an AAE>1.6 should confirm the presence of non BC absorbing material, however an AAE<1.6 does not exclude its presence. Liu et al. (2018) Another study showed that the AAE of coated BC is highly sensitive to particle size distribution and demonstrated that AAE decreases as particle size increases (Liu et al., 2018). The same study demonstrated the importance of various parameters on the BC AAE and the potential problems introduced by assuming BC AAE as being equal to 1.0. Based on the relatively large aged particles present in FinokaliaThe AAE of coated BC is highly sensitive to particle size distribution and it decreases as particle size increases (Liu et al., 2018). This last study shows that relatively low values of AAE should be expected. Therefore the relatively low average AAE does not preclude the presence of some absorbing organic aerosol. in a site like Finokal ia and demonstrated the importance of various parameters on the BC AAE and the potential problems introduced by assuming BC AAE as being equal to 1.0.

Mie theory calculations were performed in order to estimate the MAC405 and the  $E_{abs}$  due to the lensing effect of the shell covering the BC core. Initially, a non-absorbing shell was assumed. The predicted MAC405 had an average value of 15.4 m2 g-1. The average  $E_{abs}$  due to lensing effect was 2.13. The predicted average MAC405 using the measured BC size distribution and assuming pure uncoated spherical particles was 7.2 m2 g-1 varying from 6.1 to 7.8 m2 g-1 during the study (hourly averages). This is consistent with the expected 5.3-9.2 m2 g-1 for BC core diameters in the Commented [SP23]: REVIEWER 1, Comment 2

Commented [AT24]: REVIEWER 2 comment 24

Commented [AT25]: Reviewer 2 comment 3

Commented [AT26]: REVIEWER 2 comment 3

| 10-350 nm size range (Bond et al. 2006). The average predicted $MAC_{405}$ was lower than the                                      |
|------------------------------------------------------------------------------------------------------------------------------------|
| average measured MAC 405 = 16.3 m 2 g -1 . There were several periods during which the measured   |
| MAC was much higher than the -difference between the measured and the predicted values with                                        |
| the differences was as high as 8.5 m 2 g -1 -(Figure 6). These discrepancies suggest the potential           |
| presence of high levels of absorbing OA during these periods. The predicted MAC405 had an                                          |
| average value of 14.1 m 2 g -1 . The average $E_{abs}$ due to lensing effect was 2.07. The average predicted |
| MAC405 was 13% lower than the measured suggesting the existence of non-refractory absorbing                                        |
| material (Figure 6). The measured MAC405 was higher than the predicted MAC405 in 47% of the                                        |
| cases, suggesting the existence of absorbing material (Figure 6).                                                                  |

**Commented [AT27]: REVIEWER 1 comment 2**

Commented [AT28]: REVIEWER 1 comment 2

Commented [AT29]: REVIEWER 2 comment 3

In the next step, the Mie theory calculations were repeated assuming an absorbing shell with a refractive index of  $n_{OA} = 1.55 + ki$ , where the imaginary part, k, was allowed to vary from 0 to 0.4. This range of k values was selected based on previous literature (Kirchstetter et al., 2004; Alexander et al., 2008; Chakrabarty et al., 2010; Chen and Bond, 2010; Saleh et al., 2014; Chakrabarty et al., 2016, Li et al., 2016; Saleh et al., 2018). One third53%Approximately half of the resulting k values during the campaign were zero, suggesting a non-absorbing shell, while the other half were two thirds were positive. More specifically, 232% of the estimated k values ranged from 0.01 to 0.1, 722% from 0.11 to 0.2, 439% from 0.21 to 0.3, and 14% of the k values ranged from 0.31 to 0.4. During these periods, the campaign average  $b_{abs,405}$  was equal to 2.4 Mm-1. BrC was estimated to lead to a 15% increase of the campaign average  $b_{abs,405}$ .

**4.3 The role of ELVOCs**

ELVOCs can exhibit substantial larger light absorption than LVOCS or SVOCs. The association of BrC with material of lower volatility away from its sources has been reported in a number of studies focusing on biomass burning (Saleh et al., 2014; Wong et al., 2019). The hypothesis that the presence of ELVOCs could explain the higher aerosol light absorption was tested. The unexplained MAC ( $\Delta$ MAC) difference of measured and predicted values was compared with the total ELVOC mass concentration. The ELVOC concentration was estimated based on the results on the volatility analysis of the two PMF factors:

[ELVOC]= 0.15 [LO-OOA] + 0.3 [MO-OOA]

The unexplained  $b_{abs,405}$  least squares fit between (the 3 -hour average values) was well correlated with the estimated d unexplained  $b_{abs,405}$  and ELVOC concentrations with anwas R2=0.76 (Figure 7). The corresponding  $R^2$  between the 3-h average The least squares fit between the 3-hour averaged  $\Delta$ MAC and the and ELVOCs was had  $R^2$ =0.665 (Figure S167). The least squares fit between the 3-hour averaged unexplained  $b_{abs,405}$  and ELVOC was  $R^2$ =0.76 (Figure 7). Theise results suggests that the ELVOCs were probably contributing to the total absorption and could explain the difference in the MAC.

Correlation of the  $\Delta$ MAC with the rBC, and with other parameters and-were lower than those for the ELVOCs. For example, the  $R^2$  with the rBC was 0.38-(rBC), with the LO-OOA 0.44 and with the MO-OOA 0.29. There was a relatively high correlation with the sulfate levels ( $R^2$ =0.59) that could be interesting as high sulfate levels in this area correspond to high aerosol acidity which has been shown to promote formation of oligomers in secondary organic aerosol.] A comparison of the difference of measured and the predicted  $b_{abs,405}$  and the ELVOC was also conducted for all the periods including those in which no BrC was present (k=0). The correlation was  $R^2$  = 0.52. During the k=0 periods the predicted  $b_{abs,405}$  was equal or higher than the measured  $b_{abs,405}$ . This difference was due to overestimation of the absorption values by the Mie theory or due to the uncertainties in the measurements of the  $bb_{abs,405}$ .

**5. Uncertainty analysis**

The analysis presented in the previous sections has been based on a series of assumptions and, as expected is affected, by measurement uncertainties. We have tested the robustness of our conclusion about the link between the unexplained absorption and ELVOC levels by repeating the analysis for several cases. We focused on the estimation of the coating thickness of the BC particles, the The calculations were repeated in order to test how the different type of uncertainties can affect the relationship of the ELVOC and the unexplained absorption. The analysis tested the assumption of the coating thickness of the shell, the assumed refractive index of black carbon, the uncertainty of the measurements by the SP2 and PAX405, and the uncertainty of the results from the PMF analysis and finally the uncertainty of the thermodenuder as well as from the TD dynamic evaporation–model. The results of the corresponding tests are summarized in the following paragraphs.

The coating thickness was recalculated applying the using LEO algorithm calculations from on the SP2 measurements. The average coating thickness calculated by this approach was Commented [AT30]: REVIEWER 1 comment 6

Commented [SP31]: REVIEWER 2, Comment 4

approximately half of that calculated using the internal mixture assumption (Fig. S11). This resulted in lower predicted absorption and therefore a larger gap between measurements and predictions assuming that the organic aerosol was not absorbing. The unexplained absorption increased by approximately 35% (from 0.24 to 0.32 Mm-1). With the new coating thickness MHE theory calculations were repeated and a new theoretical MAC405 and babs405 were found. The calculations indicated showed that 71% of the samples had a *k*>0 suggesting an absorbing shell. The correlation between the unexplained absorption and the ELVOC concentration remained high least square fit between the averaged  $\Delta$ MAC and ELVOC was R2=0.51 while the one for the Ababs and ELVOC was R2=0.69 (Fig. S17), suggesting that despite the uncertainty in the coating thickness our results are quite robust.

The effect of the assumed BC refractive index on our results was tested by repeating the Mie theory calculations for two additional values: a relatively high value of 1.95 + 0.679i and a relatively low value 1.5 + 0.5i were assumed (Bond and Bergstrom, 2006). For the high refractive index, the predicted average MAC405 using the measured BC size distribution and assuming pure uncoated spherical particles was  $7.7 \text{ m}^2 \text{ g}^{-1}$  varying from  $6.3 \text{ to } 8.2 \text{ m}^2 \text{ g}^{-1}$  during the study (hourly averages). Even if the unexplained absorption was reduced its R2 with the ELVOCs remained high and equal to 0.72 (Fig. S18). For the low refractive index, the predicted average MAC405 of the uncoated BC particles was  $6.2 \text{ m}^2 \text{ g}^{-1}$  (range of  $5.2-6.5 \text{ m}^2 \text{ g}^{-1}$  during the study). The R2 between the unexplained absorption and the abundance of ELVOCs is quite robust with respect to the assumed value of the BC refractive index.

We estimated that 92-97% of the BC mass concentration was inside the SP2 measurement window therefore given that we also corrected for it fitting the measured size distribution, the uncertainty introduced by this limitation of the SP2 was minor. Lack et al. (2012) estimated an uncertainty of the aerosol absorption measurement babs,405 measurements by the PAX of less than 10%. Both of these uncertainties did not have an important effect on the link between the unexplained absorption and the ELVOC concentrations.

In addition, Mie theory calculations were repeated for more refractive indexes of the BC core. A maximum a refractive index  $n_{rBC,max} = 2.0 + 1.0i$  and a minimum  $n_{rBC,min} = 1.5 + 0.5i$  were assumed (Bond and Bergstrom, 2006). For the maximum refractive index, the calculations showed that 17% of the samples had a k > 0 suggesting an absorbing shell. The least square fit between the

Commented [SP32]: REVIEWER 2 Comments 9 and 11

Commented [SP33]: REVIEWER 2, Comment 6

Commented [SP34]: REVIEWER 2 Comment 7

averaged  $\Delta$ MAC and ELVOC was  $R^2$ =0.49 while the one for the  $\Delta b_{abs}$  and ELVOC was  $R^2$ =0.71. For the minimum refractive index, the calculations showed that 95 % of the samples had a k>0suggesting an absorbing shell. The least square fit between the averaged  $\Delta$ MAC and ELVOC was  $R^2$ =0.31 while the one for the  $\Delta b_{abs}$  and ELVOC was  $R^2$ =0.55.

As mentioned in a previous section the estimated measurement uncertainty of the SP2 is 40%. In addition. The PAX has an uncertainty of less than 10% for the aerosol absorption measurement  $b_{absr405}$  (Lack et al. 2012). The average propagation uncertainty of the MAC405, based on the measurements of rBC and  $b_{abs}$  at  $\lambda$ =405 nm, is 20%. An upper and lower measured MAC405 was calculated using this propagated uncertainty. For the upper limit of the measured MAC405, 89% of the samples had an absorbing shell. The relationship of the ELVOCs with the AMAC was  $R^2$ =0.23, and the relationship with the  $\Delta b_{abs}$  was  $R^2$ =0.47. Furthermore, for the lower limit of the measured MAC405 only 15% of the samples had an absorbing shell. The relationship of the ELVOCs with the AMAC was  $R^2$ =0.49, and the relationship with the  $\Delta b_{abs}$  was  $R^2$ =0.75.

An assessment of the uncertainty of the concentrations of the two OA- factors determined by the PMF analysis was performed<del>took place</del> by bootstrapping 10 runs-simulations (Ulbrich et al., 2009). The results showed that the statistical uncertainties were small indicating the accuracy of the two factors (Figure S17). The estimated uncertainty for the LO-OOA concentrations was 2% and for the MO-OOA concentrations was 3% (Fig. S20). These relatively small uncertainties suggest that the PMF uncertainty regarding the determination of these factors does not affect significantly<del>t</del> the conclusions of this study. As I was stated in a previous section, both MO-OOA and LO-OOA were positively correlated with rBC with the corresponding correlation coefficient  $R^2$  being 0.17 for MO-OOA and 0.12 for LO-OOA. These correlations between the absolute concentrations of the various PM components are in general expected in remote sites that are affected sometimes by more polluted air masses (leading to increases of the major PM components) and cleaner air masses processed by rain (leading to decreases of the major PM components. The  $R^2$  of the non-refractory PM1 and the rBC was 0.3 (Figure S18).

The uncertainty related to the volatility distributions determined by the thermodenuder results was assessed by estimating low and high limits of the ELVOC concentrations: In addition, the ELVOClow and ELVOChigh. These-concentrations were estimated based on the extreme mass fractions that were calculated duringin the sensitivity analysis of the TD model (Supplemental information Section S2). The LO-OOA ELVOC mass fractions ranged from 0.09 to (AHvap= 50 kJ

Commented [SP35]: REVIEWER 2 Comment 5

 $\frac{\text{mol}^{+}-\text{and }a_{m}=0.19)}{\text{and }0.25}$  while that of the MO-OOA from 0.13 to 0.47. For the low ELVOC case we thus assumed that:  $\frac{(\Delta H_{wap}=100 \text{ kJ mol}^{+}-\text{and }a_{m}=0.02)}{(Table 2)}$ . Similarly, the MO-OOA ELVOC mass fractions ranged from 0.13 ( $\Delta H_{wap}=50 \text{ kJ mol}^{+}-\text{and }a_{m}=0.59$ ) and 0.47 ( $\Delta H_{wap}=150 \text{ kJ mol}^{+}-\text{and }a_{m}=0.59$ ) and 0.47 ( $\Delta H_{wap}=150 \text{ kJ mol}^{+}-\text{and }a_{m}=0.09$ ) (Table 1).

For the lowest case:

**[ELVOClow]= 0.09 [LO-OOA] + 0.13 [MO-OOA]**

The R2 between the 1

The least square fit between the 3 hour averaged  $\Delta$ MAC and ELVOClow was  $R^2$ =0.68. Similarly, the least squares fit between the 3 hour averaged unexplained  $b_{abs,405}$  and ELVOClow was  $R^2$ =0.79 (Fig. S21).

For the high ELVOCest case:

**[ELVOChigh]= 0.25 [LO-OOA] + 0.47 [MO-OOA]**

Once more the correlation between the unexplained absorption at 405 nm and the ELVOCs was guite high with  $R^2=0.78$  (Fig. [S22]).

The least square fit between the 3-hour averaged AMAC and ELVOChigh was  $R^2$ =0.66. Similarly, 
[revised manuscript text omitted]

---

## Author Response (AR2)

**Response to the Comments of the Reviewer**

**(1)** The authors have done a strong job of revising their manuscript; I commend them on their work and apologize for my extreme tardiness in getting this assessment completed. I suggest that this should be publishable once the authors address remaining comments not addressed, or raised through the revisions.

We have done our best to address the comments and suggestions of the reviewer. Our responses and changes to the manuscript (in black) follow each comment of the reviewer (in blue).

**(2)** As a general thought, I would encourage the authors to consider the sensitivity of their results to the one very clear "outlier" high absorption point in their unexplained absorption vs. ELVOC concentration plots. This point is undoubtedly having an outsized effect on the fits, especially the $R^2$ value. This is not critical, but perhaps of interest. Additional comments follow.

This period characterized by high unexplained absorption and high ELVOC concentration is indeed important for the results. We do not think that though that this period should be described as an outlier. We have repeated the analysis excluding this period and the $R^2$ between unexplained absorption and ELVOC levels is lower, but still relatively high at 0.41. We have added this comment in the revised manuscript.

**(3)** The authors added mention of factors that can affect the MAC of pure BC to the intro; I suggest the authors support these contentions with references that specifically demonstrate from observations (not calculations) that the MAC is size dependent. This is certainly true if spherical particle Mie theory is used, but it is less clear this is the case for fractal particles; for example, RDG theory yields a size-independent MAC. Also, just a note that neither RDG nor Mie rely on knowledge of particle morphology; the former requires only the size of the primary spherule, but not the morphology (page 2)

We have followed the reviewer's suggestion and added references to the experimental work of Radney et al. (2014), Dastanpour et al. (2017), and Forestieri et al. (2018) together with a brief discussion. We have also rephrased the text to clarify what is needed by the RDG and the Mie theory.

**(4)** L223: The authors report a positive correlation for an R=0.31, corresponding to an $R^2 = 0.096$. This is hardly a positive correlation. I suggest that the authors perform a significance test to determine whether their correlation coefficient is significantly different from zero at some level. I am not convinced that the authors can really conclude that "the measurements suggested a positive correlation", or that the R value reported is not strongly influenced by the two outliers at low NRPM/rBC values. The same goes for the small $R^2$ values reported on L275/276.

We have performed a significance test and confirmed that these results are statistically significant at the 5% level. This information has been added to the paper.

**(5)** It would be very useful for future reference if the authors were to report the SSA value they measured for the fullerene soot. Also, the authors cite Lack et al. (2012) for the PAX uncertainty (page 5). However, Lack et al. did not use a PAX, they used a CRD-PAS and calibrated in a totally different manner than by which the PAX is calibrated. It is not clear that the uncertainties are translatable; I think the authors need to argue that the uncertainties translate between these different instruments that were calibrated in different ways and have very different sensitivities.

This is a valid point. We have rewritten this discussion of the uncertainty of the PAX measurements based, this time, on the work of Nakayama et al. (2015) who characterized the uncertainty of a Photoacoustic Extinctiometer (PAX) and a three wavelength Photoacoustic Soot Spectrometer. The measurement uncertainties for the PAX were less than 10%. The SSA value measured for the fullerene has been added to the text.

**(6)** L239: I suggest the authors provide more details regarding how they determined the values reported in Fig. S11 from the LEO method. LEO gives a result for every particle. They report an average value. But is the distribution normally distributed? And, for what size particle were these coatings determined? There are some important aspects to the LEO method with respect to how it is applied across the size distribution owing to differences in the smallest size particle measured by incandescence versus scattering in the SP2. Are these coating thicknesses from a particular size range?

We estimated the coating thickness for every BC particle with the LEO method. The average coating thickness for each period shown in Fig. S11 corresponds to the mean BC particle size ($\pm$ 5 nm) for the same period. This information has been added to the revised paper.

**(7)** Additionally, I have some concerns with the real discussion of the LEO method being left to the "uncertainty" analysis. We know for a fact that the NRPM/BC overestimates the coating thickness. Considering the LEO method is not just looking at "uncertainty;" it is arguably a fundamentally better estimate (despite the limitations of the method). This does not seem so much like an "uncertainty analysis" to me, as it is an alternative consideration. I strongly suggest integrating the LEO results with the NRPM/BC results throughout. It should be reasonably straightforward to show both the NRPM/BC and LEO-derived results together in the main text figures and to discuss them together. As a general point, I suggest that a better title than "Uncertainty analysis" would be "sensitivity analysis." The vast majority of the aspects considered in this section are assessing sensitivities of the results to methodological assumptions, which is not the same as an uncertainty assessment. It is more of a robustness assessment. Regardless, this is a welcome addition to the paper.

We agree with the point of the reviewer. We have moved the analysis using the LEO approach for the calculation of the coating thickness to Sections 4.2 and 4.3. We have also renamed the new section to "Sensitivity analysis".

**(8)** I appreciate the authors' addition of the $R^2$ values as a function of wavelength for absorption vs. BC. I suggest changing "rather inconclusive" simply to "inconclusive." I'll note also that they report only in the supplemental but not the main text the value at the longest wavelength, for which the $R^2$ is quite a bit lower than at higher wavelengths.

We have changed the characterization to "inconclusive" as suggested by the reviewer. The $R^2$ between the high wavelength $b_{abs}$ ($\lambda$=950 nm) and the rBC mass has been added to the main text.

**(9)** P14: I suggest that the authors in their first sentence change it to "between the measured and the reference value" rather than "theoretical value," in line with my previous concern that it appeared the authors were comparing their observations with a calculated ("theoretical") value. The Bond et al. value is not theoretical—it is explicitly derived from experiments.

We have rephrased this sentence to avoid confusion and we now discuss the "reference value".

**(10)** P15: The average predicted MAC405 is likely not lower than the measured value once uncertainties are accounted for. This should be noted. The authors should provide statistical evidence that the predicted value is lower, based on e.g. a t-test or some other appropriate test. We have followed the suggestion of the reviewer and analyzed the statistical significance of the difference between the average measured and estimated value. The MAC uncertainty due to the corrections introduced by the size limitation of the SP2 was minor and that by the PAX measurements was less than 10%. Based on these an uncertainty of the measured $MAC_{405}$ approximately equal to 10% has been estimated. The t-test suggested a p=0.078 for the predicted value being less than the measured one. These points have been added in the revised manuscript.

**(11)** P14: I also suggest that the authors modify their discussion in relation to the Gyawali et al. (2009) study to indicate that those were theoretical values obtained via spherical particle Mie theory, which might not yield (and probably does not yield) the correct size dependence—which is what gives rise to the large variation in the AAE with size in that paper and others that have followed. For the Lui et al. study cited, the authors here should not just note that the AAE decreases with size (for more complex calculations) but that it decreases to only ~0.8 at the lower end for their realistic calculations. The authors should be more specific than saying "relatively low values of AAE should be expected." How low is low? If 0.8 for collapsed BC is expected, then an observed value of 1 still indicates a relatively small contribution of brown carbon.
We have added in the revised paper the information suggested by the reviewer regarding the Gyawali et al. (2009) and Lui et al. (2018) studies. According to the results of Gyawali et al. (2009), the average AAE for the present study based on the core diameter measured and the estimated coating thickness can range from 1 to 1.25. This has also been added in the revised version.

**(12)** P15: I find the statement that "The association of BrC with material of lower volatility away from its sources has been reported in a number of studies focusing on biomass burning (Saleh et al., 2014; Wong et al., 2019)," unclear. The Saleh study is a laboratory study on biomass burning particles; it is by definition right by the source. But, they did consider the influence of simulated photochemical aging. The Wong et al. study is also a laboratory study, but one that did consider photo-aging of BrC. If the authors here mean that the Saleh and Wong et al. photochemically aged their laboratory samples and this means "away from its sources" I suggest they be more specific.
This is a valid point. We have rephrased this sentence replacing the statement "away from its sources" with "photochemical aged aerosol" to be more precise.

**(13)** P17: There is a typo: the RI should be + 0.79i, not 0.679i.
The typo has been corrected.

**(14)** Data Availability: In line with current best practice and expectations for articles published in ACP, I suggest that the authors archive their data from the publicly funded study in an accessible archive prior to the publication of this work.
A significant fraction of the result of this field study has not been published yet. A couple of papers are under preparation. All the results will be archived in an accessible repository after the completion of the publication process. In the meantime, the results presented in the paper are archived in a restricted repository and are available upon request.

**(15)** Presentation of Literature Results: Upon reading again, I have a few concerns over how the previous literature is presented. Specific comments follow here:
L38: The authors should clarify that the measured Eabs accounts for the combined effects of lensing and BrC. This would help set up the discussion that follows better, especially when they report measurements of Eabs from different studies without clearly delineating the relative contributions of lensing and BrC.

We now clarify in the revised paper that the measured Eabs accounts for the combined effects of both lensing and BrC.

**(16)** L49: I suggest it would be fairer to say that studies have found a wide range of effects.

The text has been rephrased following the reviewer's suggestion.

**(17)** L50: The authors misstate the results of Liu. Here, the authors report the results from the TD, but Liu et al emphasize that the MAC-based method is superior. Their 405 nm value is larger than the 781 nm value, when viewed this way. And this is largely a result of BrC absorption, with lensing making some contribution. I suggest this be revised to reflect the results of Liu et al. more accurately.

Additional information about the findings and conclusions of Liu et al. (2005) has been added to the text.

**(18)** General comment on presentation of literature results: In general, I suggest the authors report the uncertainties associated with each of the specific Eabs values they mention. There can be a wide variability/range of values for a given mean value. Or, they should clarify that these are means. I also suggest that they introduce greater consistency in terms of attributing different effects to lensing versus brown carbon. For example, for the Zhang et al. 2018a study, they mention two wavelengths with no mention of attribution. But for the Lack et al. (2012) study, they mention contributions of BrC.

We do clarify now that the reported Eabs values are averages of a range of values. The variability is also reported when available. We have tried to improve the consistency of the reported effects in the literature review. Unfortunately, we are limited by the analysis performed in each study. For example, in a number of studies there is no effort to quantify the contribution of BrC.

**(19)** For the Liu et al. (2017) study, the authors might clarify that above Rbc = 3 the enhancement is not only "significant," but that they observed a continual variation in Eabs with increasing Rbc.

We have added the suggested clarification to the revised paper.

**(20)** For the Cappa et al. (2012) study, I believe they also report values using the MAC method, and find results similar to the TD method.

The above information has been added.

**(21)** L74: Hearly should be Healy.

The typo has been corrected.

**(22)** Rbc: The authors should clarify that different methods are used to determine Rbc. For Liu et al. (2017), for example, it is based on single particle SP2 measurements. But for most other studies mentioned, the Rbc (if measured at all) is an ensemble average.
We have added the requested information about the method used to measure the $R_{BC}$.

**(23)** For McMeeking et al. (2014), the absorption enhancement was negligible up to Rbc = 10. Not just below Rbc = 1.
The suggested clarification has been added.

**(24)** L59: cite should be site.
The text has been corrected.

**(25)** L137: The distributions were almost certainly not fit with a Gaussian. They were fit with a log-normal distribution. (If they were actually fit with a Gaussian, this needs to be justified.)
The data were fitted using a log normal distribution. The text has been corrected.

[revised manuscript text omitted]

Previous studies have demonstrated that lensing has a wide range of strong effects on the
light absorption of BC. Liu et al. (2015) quantifiedreported the $E_{abs}$ in a rural area near London
during the winter. They used two methods to find the $E_{abs}$. Using the TD method they found a
campaign--average $E_{abs}$ of 1.3 at wavelength ($\lambda$) equal to 405 nm and 1.4 at 781 nm. They also
estimated the In $E_{ab}$ the second method they combiningpared their measurements with a reference
MAC from the literature to estimate the $E_{abs}$,m in a rural area near London during the winter. At
the large wavelength the two methods agreed, but this was not the case at the lower but at the
small wavelength the difference was significantly higher. They argued that the $E_{abs}$ calculated
using the reference MAC was more accurate. In addition, they showed that there was is a

[revised manuscript text omitted]

**5.  Sensitivity analysis**

The analysis presented in the previous sections has been based on a series of assumptions and, as expected is affected, by measurement uncertainties. We have tested the robustness of our conclusion about the link between the unexplained absorption and ELVOC levels by repeating the analysis for several cases. We focused on the estimation of the coating thickness of the BC particles, the refractive index of black carbon, the uncertainty of the measurements by the SP2 and PAX$_{405}$, the uncertainty of the results from the PMF analysis and finally the uncertainty of the thermodenuder model. The results of the corresponding tests are summarized in the following paragraphs.

The coating thickness was recalculated applying the LEO algorithm on the SP2 measurements. The average coating thickness calculated by this approach was approximately half of that calculated using the internal mixture assumption (Fig. S11). This resulted in lower predicted absorption and therefore a larger gap between measurements and predictions assuming that the organic aerosol was not absorbing. The unexplained absorption increased by approximately 35% (from 0.24 to 0.32 Mm$^{-1}$). The calculations indicated that 71% of the samples had a $k>0$ suggesting an absorbing shell. 
[revised manuscript text omitted]